# POSTERIOR SAMPLING MODEL-BASED POLICY OPTIMIZATION UNDER APPROXIMATE INFERENCE

## ABSTRACT

Model-based reinforcement learning algorithms (MBRL) hold tremendous promise for improving the sample efficiency in online RL. However, many existing popular MBRL algorithms cannot deal with exploration and exploitation properly. Posterior sampling reinforcement learning (PSRL) serves as a promising approach for automatically trading off the exploration and exploitation, but the theoretical guarantees only hold under exact inference. In this paper, we show that adopting the same methodology as in exact PSRL can be suboptimal under approximate inference. Motivated by the analysis, we propose an improved factorization for the posterior distribution of polices by removing the conditional independence between the policy and data given the model. By adopting such a posterior factorization, we further propose a general algorithmic framework for PSRL under approximate inference and a practical instantiation of it. Empirically, our algorithm can surpass baseline methods by a significant margin on both dense rewards and sparse rewards tasks from the Deepmind control suite, OpenAI Gym and Metaworld benchmarks.

## 1 INTRODUCTION

Model-based reinforcement learning has demonstrated great success in improving the sample efficiency of RL. However, many existing popular model-based algorithms (Kurutach et al., 2018; Chua et al., 2018; Janner et al., 2019) cannot deal with exploration and exploitation properly, and hence may lead to poor performance when exploration is crucial. To trade off exploration and exploitation, most of the existing algorithms can be categorized into 1) *optimism-based* (Jaksch et al., 2010; Pacchiano et al., 2021; Curi et al., 2020); 2) *posterior-sampling-based* (Strens, 2000; Osband et al., 2013; 2018; Fan & Ming, 2021); and 3) *information-directed sampling* (Russo & Van Roy, 2014) approaches.

As shown by Osband & Van Roy (2017), posterior sampling reinforcement learning (PSRL) can match the statistical efficiency (or regret bound) of optimism-based algorithms, but enjoys better computational efficiency. Information-directed sampling methods can be more statistically efficient when faced with complex information structure (Russo & Van Roy, 2014), but require estimators for the mutual information, which is difficult for high-dimensional random variables. Hence, in this paper, we focus on posterior sampling. For simplicity, we restrict attention to the episodic RL setting.

Under the PSRL framework, one maintains a posterior $p(\mathcal{M}|\mathcal{D}_\mathcal{E})$ of the Markov decision process (MDP) $\mathcal{M}$ given the observations $\mathcal{D}_\mathcal{E}$ collected in real environment $\mathcal{E}$. At the beginning of each episode, an MDP is sampled from the posterior, and then we compute the optimal policy $\pi(\mathcal{M})$ for the sampled model $\mathcal{M}$. Equivalently, we can also view this policy as a sample from a "degenerate" posterior[1] over policies of the form $p(\pi|\mathcal{D}_\mathcal{E}) = \int \delta(\pi|\mathcal{M})p(\mathcal{M}|\mathcal{D}_\mathcal{E})d\mathcal{M}$, where $\delta(\pi|\mathcal{M}) = \delta(\pi - \pi(\mathcal{M}))$ is a Dirac delta distribution. This policy is then executed in the real environment to collect new data. Theoretically, such a simple strategy has been shown to achieve a Bayesian regret of $\tilde{\mathcal{O}}(\sqrt{K})$ for $K$ episodes (Osband et al., 2013; Osband & Van Roy, 2014). However, the theoretical guarantees only hold under exact inference, i.e., when we have access to the true posterior over models $p(\mathcal{M}|\mathcal{D}_\mathcal{E})$, and when we can compute the optimal policy, which is very unlikely in practice.

A common heuristic approximation to PSRL (see e.g., Fan & Ming (2021)) is to replace the posterior over models with some approximation, such as Bayesian linear regression on top of the representations

---

[1]For simplicity, we assume that there is only one optimal policy for each MDP.

learned by a neural network. In this way, the resulting policy is then sampled from $q^\delta(\pi|\mathcal{D}_\mathcal{E}) = \int \delta(\pi|\mathcal{M})q(\mathcal{M}|\mathcal{D}_\mathcal{E})d\mathcal{M}$, where we replace $p(\mathcal{M}|\mathcal{D}_\mathcal{E})$ with the approximate posterior $q(\mathcal{M}|\mathcal{D}_\mathcal{E})$.

At first glance, such a heuristic choice is natural as it shares the same form as the true posterior $p(\pi|\mathcal{D}_\mathcal{E})$. However, in this paper, we prove that we can get lower regret if we replace the degenerate $\delta(\pi|\mathcal{M})$ with a non-degenerate distribution of the form $q(\pi|\mathcal{M}, \mathcal{D}_\mathcal{E})$, that depends on the model as well as empirical data $\mathcal{D}_\mathcal{E}$; this is necessary to compensate for the fact that the posterior over models $q(\mathcal{M}|\mathcal{D}_\mathcal{E})$ may be suboptimal. By tuning the relative strength of the dependence of $\pi$ on $\mathcal{D}_\mathcal{E}$ and $\mathcal{M}$, we can find a sweet spot between maximizing the data efficiency and minimizing the effect of approximate inference error. Furthermore, such a decomposition is guaranteed to be no worse than the standard approach of using $q^\delta(\pi|\mathcal{D}_\mathcal{E})$.

Building upon the above results, we come up with a generic framework for PSRL under approximate inference. To implement the method in practice, we combine deep ensembles (Lakshminarayanan et al., 2017) and Model-based Policy Optimization (MBPO) (Janner et al., 2019). We also propose two different sampling strategies for policy selection that exploit our posterior approximation. Empirically, our algorithm significantly outperforms the baselines on both dense reward and sparse reward tasks (Brockman et al., 2016; Tunyasuvunakool et al., 2020; Yu et al., 2020). Additionally, we also conduct various ablation studies to provide a better understanding of our algorithm.

In summary, our contributions are

1. We conduct a rigorous study on how approximate inference affects the Bayesian regret in PSRL, showing that adopting the same methodology as in exact PSRL may be suboptimal when the true posterior is unavailable (Section 2).
2. Motivated by our analysis, we develop a generic framework for PSRL under approximate inference as well as a practical version of it based on deep ensembles and (optimistic) sampling approaches for the policies (Section 3).
3. We present empirical results on DM control suite, OpenAI Gym and Metaworld benchmarks to demonstrate the efficacy of the proposed approach (Section 4).

## 2 PROBLEM STATEMENT AND THEORETICAL RESULTS

We start by introducing some notation, and summarizing prior work, before presenting our new theoretical results, which forms a basis for our algorithm.

**Notation.** We consider the finite-horizon episodic Markov Decision Process (MDP) problem, of which we denote an instance as $\mathcal{M} := \{\mathcal{S}, \mathcal{A}, r_\mathcal{M}, p_\mathcal{M}, H, \rho\}$. For each instance $\mathcal{M}$, $\mathcal{S}$ and $\mathcal{A}$ denote the set of states and actions, respectively. $r_\mathcal{M} : \mathcal{S} \times \mathcal{A} \to [0, R_{\max}]$ is the reward function, $p_\mathcal{M}$ is the transition distribution, $H$ is the length of the episode, and $\rho$ is the distribution of the initial state. We further define the value function of a policy $\pi$ under MDP $\mathcal{M}$ at timestep $i$ as

$$V_{\pi,i}^\mathcal{M}(\boldsymbol{s}) := \mathbb{E}_{\mathcal{M},\pi}\left[\sum_{t=i}^H r_\mathcal{M}(\boldsymbol{s}_t, \boldsymbol{a}_t)\,|\,\boldsymbol{s}_i = \boldsymbol{s}\right], \tag{1}$$

where $\boldsymbol{s}_{t+1} \sim p_\mathcal{M}(\boldsymbol{s}|\boldsymbol{s}_t, \boldsymbol{a}_t)$ and $\boldsymbol{a}_t \sim \pi(\boldsymbol{a}|\boldsymbol{s}_t)$. We define $\pi^\star$ as the optimal policy for an MDP $\mathcal{M}$ if $V_{\pi^\star,i}^\mathcal{M}(\boldsymbol{s}) = \max_\pi V_{\pi,i}^\mathcal{M}(\boldsymbol{s})$ for all $\boldsymbol{s} \in \mathcal{S}$ and $i \in [1, H]$. We define the cumulative reward obtained by policy $\pi$ over $H$ steps sampled from model $\mathcal{M}$ as follows:

$$R_\mathcal{M}(\pi) = \mathbb{E}_{\mathcal{M},\pi}\left[\sum_{t=1}^H r_\mathcal{M}(\boldsymbol{s}_t, \boldsymbol{a}_t)\right] \quad \text{where} \quad \boldsymbol{s}_{t+1} \sim p_\mathcal{M}(\boldsymbol{s}|\boldsymbol{s}_t, \boldsymbol{a}_t) \text{ and } \boldsymbol{a}_t \sim \pi(\boldsymbol{a}|\boldsymbol{s}_t), \tag{2}$$

where the initial state is sampled from $\boldsymbol{s}_1 \sim \rho(\boldsymbol{s})$.

**Regret.** For a given MDP $\mathcal{M}$, the regret is defined as the difference between value function of the optimal policy in hindsight and that of the actual policy executed by the algorithm $\mathscr{A}$,

$$\text{Regret}(T, \mathscr{A}, \mathcal{M}) := \sum_{k=1}^K \underbrace{\int \rho(\boldsymbol{s}_1)\left(V_{\pi^\star,1}^\mathcal{M}(\boldsymbol{s}_1) - V_{\pi^k,1}^\mathcal{M}(\boldsymbol{s}_1)\right)d\boldsymbol{s}_1}_{:=\Delta_k}, \tag{3}$$

where $\pi^\star$ is the optimal policy for $\mathcal{M}$, and $\pi^k$ is the policy employed by the algorithm for $k_{th}$ episode. Correspondingly, the Bayesian regret is defined as the expectation of the above regret, i.e.,

$$\text{BayesianRegret}(T, \mathscr{A}, p(\mathcal{M})) := \mathbb{E}\left[\text{Regret}(T, \mathscr{A}, \mathcal{M})\right] = \mathbb{E}\left[\sum_{k=1}^{K} \Delta_k\right]. \tag{4}$$

Here the expectation is taken over the prior distribution of dynamics models $p(\mathcal{M})$ and the randomness in the algorithm $\mathscr{A}$ and environment.

**PSRL.** Posterior Sampling Reinforcement Learning or PSRL (Strens, 2000) serves as a generic algorithmic framework for automatically trading off exploration and exploitation in online RL. The core of PSRL is computing the posterior over MDPs (dynamics and reward models). Conditioned on the data $\mathcal{D}_\mathcal{E}$ collected from the environment, we denote the posterior of the model as $p(\mathcal{M}|\mathcal{D}_\mathcal{E})$. Hence, the posterior distribution of policies is

$$p(\pi|\mathcal{D}_\mathcal{E}) = \int p(\pi|\mathcal{M})\, p(\mathcal{M}|\mathcal{D}_\mathcal{E}) d\mathcal{M}, \quad \text{where } p(\pi|\mathcal{M}) = \delta(\pi|\mathcal{M}) \tag{5}$$

where $\delta(\pi|\mathcal{M})$ is a Dirac delta distribution, defined as $\delta(\pi(\mathcal{M})|\mathcal{M}) = 1$, where $\pi(\mathcal{M}) = \arg\max_\pi R_\mathcal{M}(\pi)$ is the optimal policy for solving the MDP $\mathcal{M}$. At the beginning of every episode, an MDP (or equivalently a policy) is sampled from the posterior distribution, and is then used for collecting new data. This simple algorithmic framework not only attains a Bayesian regret of $\tilde{O}(\sqrt{K})$ (Osband et al., 2013), where $K$ is the total number of episodes, but also enjoys better empirical performance than optimism-based methods in bandits (Chapelle & Li, 2011; Osband & Van Roy, 2017). Nevertheless, the theoretical results only hold under exact inference. In the rest of this section, we discuss how performance is affected when we use approximate inference.

**Bayesian regret under approximate inference.** Let us denote the approximate and true posterior distribution of polices at $k_{th}$ episode by

$$q_k(\pi) = q(\pi|\mathcal{D}_\mathcal{E}^k) \quad \text{and} \quad p_k(\pi) = \int \delta(\pi|\mathcal{M}) p(\mathcal{M}|\mathcal{D}_\mathcal{E}^k) d\mathcal{M}. \tag{6}$$

where $\mathcal{D}_\mathcal{E}^k$ denotes all the data collected from the environment $\mathcal{E}$ till the $k$-th episode . Next we characterize how approximate posterior inference affects the Bayesian regret.

**Theorem 1** *For $K$ episodes, the Bayesian regret of posterior sampling reinforcement learning algorithm $\mathscr{A}$ with any approximate posterior distribution $q_k$ at episode $k$ is upper bounded by*

$$\sqrt{CK(HR_{\max})^2 \mathbb{H}(\pi^\star)} + 2HR_{\max} \sum_{k=1}^{K} \sqrt{\mathbb{E}\left[\mathbf{d}_{KL}\left(q_k(\pi)\,|\,p_k(\pi)\right)\right]}, \tag{7}$$

*where $\mathbb{H}(\pi^\star)$ is the entropy of the prior distribution of optimal polices, i.e., $p(\pi) = \int \delta(\pi|\mathcal{M}) p(\mathcal{M}) d\mathcal{M}$, $C$ is some problem-dependent constant and $\mathbf{d}_{KL}(\cdot|\cdot)$ is the KL-divergence.*

Theorem 1 (proved in the appendix) provides an upper bound of the Bayesian regret under approximate posterior inference. The first term is $\tilde{\mathcal{O}}(\sqrt{K})$. The second term will be zero under exact posterior inference. However, when performing approximate inference, we see that we should choose the approximate posterior distribution $q(\pi|\mathcal{D}_\mathcal{E})$ as "close" to the true distribution $p(\pi|\mathcal{D}_\mathcal{E})$ as possible. A natural and common choice of $q(\pi|\mathcal{D}_\mathcal{E})$ would be the following one, which takes the same form as the true posterior,

$$q^\delta(\pi|\mathcal{D}_\mathcal{E}) := \int \delta(\pi|\mathcal{M}) q(\mathcal{M}|\mathcal{D}_\mathcal{E}) d\mathcal{M}. \tag{8}$$

However, the following proposition shows that such a choice is suboptimal.

**Proposition 1** *Under approximate inference (i.e., $q(\mathcal{M}|\mathcal{D}_\mathcal{E}) \neq p(\mathcal{M}|\mathcal{D}_\mathcal{E})$), the optimal $q(\pi|\mathcal{M})$ may not be a Dirac delta distribution, i.e., there exists other $q(\pi|\mathcal{D}_\mathcal{E})$ such that*

$$\mathbf{d}_{KL}\left(q(\pi|\mathcal{D}_\mathcal{E})\,|\,p(\pi|\mathcal{D}_\mathcal{E})\right) \leq \mathbf{d}_{KL}\left(q^\delta(\pi|\mathcal{D}_\mathcal{E})\,|\,p(\pi|\mathcal{D}_\mathcal{E})\right). \tag{9}$$

As an illustration, we provide the following example, which also serves as a constructive proof of Proposition 1.

> EXAMPLE 1. SUBOPTIMALITY OF $q^\delta(\pi|\mathcal{M})$.
>
> Consider a toy setting, where the support set of MDPs is $\{\mathcal{M}_1, \mathcal{M}_2\}$, and the support set of policies is $\{\pi_1, \pi_2\}$. Suppose that the true posterior distribution of MDPs is $p(\mathcal{M}_1|\mathcal{D}_\mathcal{E}) = 1/3$, $p(\mathcal{M}_2|\mathcal{D}_\mathcal{E}) = 2/3$, and the optimal policy per MDP is $\delta(\pi_1|\mathcal{M}_1) = 1$ and $\delta(\pi_2|\mathcal{M}_2) = 1$. This we get the following exact distribution over policies: $p(\pi|\mathcal{D}_\mathcal{E})$ is
>
> $$p(\pi|\mathcal{D}_\mathcal{E}) = \underbrace{\begin{bmatrix} \delta(\pi_1|\mathcal{M}_1)=1, \ \delta(\pi_1|\mathcal{M}_2)=0 \\ \delta(\pi_2|\mathcal{M}_1)=0, \ \delta(\pi_2|\mathcal{M}_2)=1 \end{bmatrix}}_{\delta(\pi|\mathcal{M})} \underbrace{\begin{bmatrix} p(\mathcal{M}_1|\mathcal{D}_\mathcal{E})=\frac{2}{3} \\ p(\mathcal{M}_2|\mathcal{D}_\mathcal{E})=\frac{1}{3} \end{bmatrix}}_{p(\mathcal{M}|\mathcal{D}_\mathcal{E})} = \begin{bmatrix} p(\pi_1|\mathcal{D}_\mathcal{E})=\frac{2}{3} \\ p(\pi_2|\mathcal{D}_\mathcal{E})=\frac{1}{3} \end{bmatrix} \qquad (10)$$
>
> Now suppose we use the approximate posterior distribution over models, $q(\mathcal{M}_1|\mathcal{D}_\mathcal{E}) = 0$ and $q(\mathcal{M}_2|\mathcal{D}_\mathcal{E}) = 1$. We can optimize $q(\pi|\mathcal{M})$ by minimizing $\mathbf{d}_{\mathrm{KL}}\left(q(\pi|\mathcal{D}_\mathcal{E}) \big| p(\pi|\mathcal{D}_\mathcal{E})\right)$. One solution could be
>
> $$q(\pi|\mathcal{D}_\mathcal{E}) = \underbrace{\begin{bmatrix} q(\pi_1|\mathcal{M}_1)=\frac{1}{2}, \ q(\pi_1|\mathcal{M}_2)=\frac{2}{3} \\ q(\pi_2|\mathcal{M}_1)=\frac{1}{2}, \ q(\pi_2|\mathcal{M}_2)=\frac{1}{3} \end{bmatrix}}_{q(\pi|\mathcal{M})} \underbrace{\begin{bmatrix} q(\mathcal{M}_1|\mathcal{D}_\mathcal{E})=0 \\ q(\mathcal{M}_2|\mathcal{D}_\mathcal{E})=1 \end{bmatrix}}_{q(\mathcal{M}|\mathcal{D}_\mathcal{E})} = \begin{bmatrix} q(\pi_1|\mathcal{D}_\mathcal{E})=\frac{2}{3} \\ q(\pi_2|\mathcal{D}_\mathcal{E})=\frac{1}{3} \end{bmatrix} \qquad (11)$$
>
> We see that the optimal $q(\pi|\mathcal{M})$ requires modeling uncertainty in the policy even conditional on the model. By contrast, if we adopt $q^\delta(\pi|\mathcal{D}_\mathcal{E})$ as our approximation, we will have
>
> $$\mathbf{d}_{\mathrm{KL}}\left(q^\delta(\pi|\mathcal{D}_\mathcal{E}) \big| p(\pi|\mathcal{D}_\mathcal{E})\right) = \log 3 = \max_{q \in \Delta^1} \mathbf{d}_{\mathrm{KL}}\left(q(\pi|\mathcal{D}_\mathcal{E}) \big| p(\pi|\mathcal{D}_\mathcal{E})\right). \qquad (12)$$

The above example tells us that the $q^\delta(\pi|\mathcal{D}_\mathcal{E})$ can perform arbitrarily poorly in terms of the KL divergence. Given this observation, a natural follow-up question would be: is there a better choice other than $q^\delta(\pi|\mathcal{D}_\mathcal{E})$? We provide an answer in the next section.

## 3 METHOD

Motivated by the results in the previous section, we first introduce a posterior decomposition which is guaranteed to be better than $q^\delta(\pi|\mathcal{D}_\mathcal{E})$. Then we introduce a practical version of the algorithm built upon deep ensembles (Lakshminarayanan et al., 2017). Finally we propose two sampling approaches for encouraging efficient exploration.

### 3.1 POSTERIOR DECOMPOSITION

In section 2, we showed that $q^\delta(\pi|\mathcal{D}_\mathcal{E})$ is not a favorable choice. This is because it assumes that, once $\mathcal{M}$ is given, the policy $\pi$ is determined. This motivates us to consider the following more flexible posterior decomposition of $q(\pi|\mathcal{D}_\mathcal{E})$,

$$q(\pi|\mathcal{D}_\mathcal{E}, \lambda) = \int q(\pi|\mathcal{M}, \mathcal{D}_\mathcal{E}, \lambda) q(\mathcal{M}|\mathcal{D}_\mathcal{E}) d\mathcal{M}. \qquad (13)$$

Intuitively, such a posterior decomposition no longer assumes that the model can capture all the relevant properties of the data. We illustrate these two posterior approximations in Figure 1. The extra parameter $\lambda \in [0, 1]$ allows us to balance the importance of fictitious data ($\mathcal{D}_\mathcal{M}$) from $\mathcal{M}$ and the real data ($\mathcal{D}_\mathcal{E}$) from the environment. In particular, we define

$$q(\pi|\mathcal{M}, \mathcal{D}_\mathcal{E}, \lambda = 0) = q(\pi|\mathcal{M}) = \delta(\pi|\mathcal{M}) \qquad (14)$$
$$q(\pi|\mathcal{M}, \mathcal{D}_\mathcal{E}, \lambda = 1) = q(\pi|\mathcal{D}_\mathcal{E}) \qquad (15)$$

Thus when $\lambda$ is small, we trust our model more, whereas when $\lambda$ is large we trust it less. In the extreme case where $\lambda = 0$, this framework reduces to the degenerate posterior $q^\delta(\pi|\mathcal{D}_\mathcal{E})$. By adjusting $\lambda$, we can find a sweet spot in minimizing the effect of approximate inference error and maximizing the data efficiency. More formally, the following proposition illustrates the advantage of equation 13.

**Proposition 2** *By adopting the posterior decomposition of equation 13, we have*

$$\min_\lambda \mathbf{d}_{KL}\left(q(\pi|\mathcal{D}_\mathcal{E}, \lambda) | p(\pi|\mathcal{D}_\mathcal{E})\right) \leq \mathbf{d}_{KL}\left(q^\delta(\pi|\mathcal{D}_\mathcal{E}) \big| p(\pi|\mathcal{D}_\mathcal{E})\right). \qquad (16)$$

**Algorithm 1** PSRL with approximate inference using **Ensemble Sampling (PS-MBPO)** or **Optimistic Ensemble Sampling (OPS-MBPO)**.

**Require:** Initialize an ensemble of dynamics models $\Theta = \{\hat{\boldsymbol{\theta}}_n\}_{n=1}^{N}$ i.i.d. $\sim q(\boldsymbol{\theta})$.
**Require:** Initialize an ensemble of policy networks $\Phi = \{\hat{\boldsymbol{\phi}}_{n,m}\}_{n,m=1}^{N,M}$ i.i.d. $\sim q(\boldsymbol{\phi})$.
**Require:** Initialize empty datasets $\mathcal{D}_{\mathcal{E}}$ and $\{\mathcal{D}_{\mathcal{M}}^{n,m}\}_{n,m=1}^{N,M}$. Real data vs. synthetic data ratio $\lambda$.
1: **for** $K$ episodes **do**
      ▷   /* Dynamics training. (Line 2) */
2:    Train the ensemble models $\Theta$ on $\mathcal{D}_{\mathcal{E}}$ under the objective in equation 17.
      ▷   /* Policy sampling. (Line 3) */
3:    Sample a policy $\pi$ from $\Phi$ **uniformly at random** (equation 18) or based on **the optimistic distribution** (equation 19).
4:    Sample state $\boldsymbol{s}_1$ from the initial state distribution $\rho(\boldsymbol{s})$
5:    **for** $h = 2 : H$ steps **do**
        ▷   /* Data collection. (Lines 6-11) */
6:        $\boldsymbol{s}_h$ = rollout(world dynamics $\mathcal{E}$, policy $\pi$, initial state $\boldsymbol{s}_{h-1}$, num. steps 1)
7:        Add $\boldsymbol{s}_h$ to $\mathcal{D}_{\mathcal{E}}$
8:        Sample state $\boldsymbol{s} \sim \mathcal{D}_{\mathcal{E}}$
9:        **for** each model $n$, policy $m$ **do**
10:          $\mathcal{D}_{\mathcal{M}}^{n,m}$ = rollout(dynamics $\hat{\boldsymbol{\theta}}_n$, policy $\hat{\boldsymbol{\phi}}_{n,m}$, initial state $\boldsymbol{s}$, num. steps $R$)
11:          Created mixed dataset $D = \lambda \mathcal{D}_{\mathcal{E}} + (1-\lambda)\mathcal{D}_{\mathcal{M}}^{n,m}$[2]
          ▷   /* Policy optimization (Line 12) */
12:          $\hat{\boldsymbol{\phi}}_{n,m}$ = update-policy($\hat{\boldsymbol{\phi}}_{n,m}$, $D$, num. gradient steps $G$)
13:        **end for**
14:    **end for**
15:    **Update the optimistic policy distribution** (equation 19).
16: **end for**

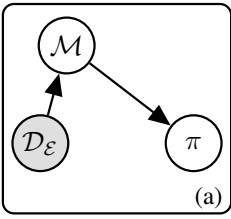
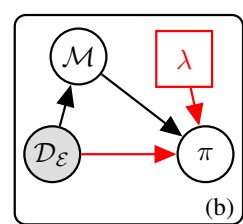
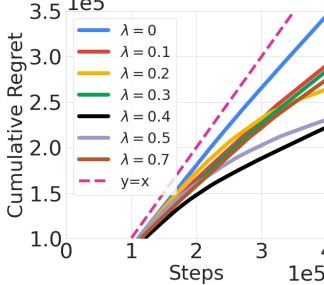

Figure 1: Graphical models for (a) the standard and (b) our posterior over policies $\pi$. Differences are shown in red.

Figure 2: A comparison of cumulative regret for different $\lambda$.

The above proposition informs us that we can minimize the KL divergence, and hence the upper bound of the Bayesian regret, by carefully choosing the value of $\lambda$. As empirical evidence, in Figure 2 we plot the cumulative regret with different $\lambda$ in the cartpole-swingup environment with sparse reward. We see that a value of about $\lambda = 0.4$ or $\lambda = 0.5$ is the best, and $\lambda = 0$ is the worst in this case.

### 3.2 PROPOSED ALGORITHM

Building upon the above observations, we introduce a simple and general algorithmic framework for PSRL under approximate inference, which only differs from the standard PSRL framework in the decomposition of policy posterior (see Algorithm 2 in Appendix C). To implement the method in practice, we use deep ensembles (Lakshminarayanan et al., 2017) to approximate the posterior distributions $q(\mathcal{M}|\mathcal{D}_{\mathcal{E}})$ and $q(\pi|\mathcal{M}, \mathcal{D}_{\mathcal{E}})$ with $\Theta$ and $\Phi$ (see Algorithm 1). This is similar to ME-TRPO (Kurutach et al., 2018), PETS (Chua et al., 2018) and MBPO (Janner et al., 2019), except we also model the uncertainty over policies (i.e., $q(\pi|\mathcal{M}, \mathcal{D}_{\mathcal{E}})$) in addition to dynamics (i.e., $q(\mathcal{M}|\mathcal{D}_{\mathcal{E}})$).

---

[2]By mixed dataset $\lambda \mathcal{D}_{\mathcal{E}} + (1-\lambda)\mathcal{D}_{\mathcal{M}}^{n,m}$, we mean that for each data point in the training batch, it is with probability of $\lambda$ being sampled from the real data $\mathcal{D}_{\mathcal{E}}$ and probability of $1-\lambda$ from the fictitious data $\mathcal{D}_{\mathcal{M}}^{n,m}$.

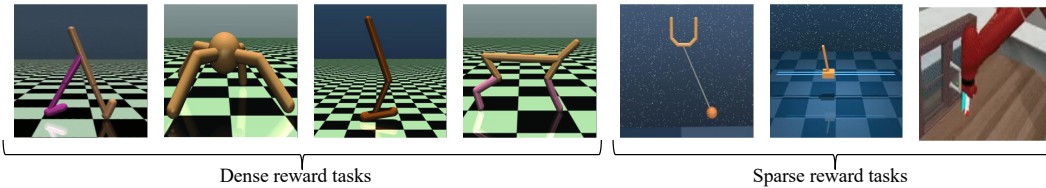

Dense reward tasks          Sparse reward tasks

Figure 3: We consider seven tasks from three benchmarks: OpenAI Gym, DM Control and Metaworld. These seven tasks cover both dense reward and sparse reward tasks.

In more detail, each member in the deep ensemble is a conditional Gaussian distribution over outputs, characterized by its mean $\boldsymbol{\mu}$ and variance $\boldsymbol{\sigma}^2$. For multi-dimensional predictions, we treat each dimension independently, and only predict the marginal mean and marginal variance for simplicity. Each ensemble member is trained independently by minimizing the negative log-likelihood,

$$-\log p_{\boldsymbol{\theta}}(y|\boldsymbol{x}) = \frac{\log \sigma_{\boldsymbol{\theta}}^2(\boldsymbol{x})}{2} + \frac{(y - \mu_{\boldsymbol{\theta}}(\boldsymbol{x}))^2}{2\sigma_{\boldsymbol{\theta}}^2(\boldsymbol{x})} + \text{constant}. \tag{17}$$

We maintain $N$ different dynamics models. For each such model, we also compute $M$ different policies; we use the soft actor-critic (SAC) (Haarnoja et al., 2018) method (see the appendix for details). The policy network $\pi_{n,m}$ is updated based on synthetic data $\mathcal{D}_{\mathcal{M}}^{n,m}$ generated by dynamics model $n$ and policy model $m$, as well as environmental data $\mathcal{D}_{\mathcal{E}}$ generated by interacting the real world dynamics with a sampled policy. See Algorithm 1 for the pseudo-code.

### 3.3 SAMPLING POLICIES

**Ensemble Sampling.** Given the posterior distributions, it remains to specify the sampling approach for policies. The simplest sampling strategy is uniform sampling at the beginning of each episode,

$$\pi \sim \mathcal{U}(\{\pi_{1,1}, ..., \pi_{N,M}\}). \tag{18}$$

In the case of bandits (where $N = 1$, since there is no transition model), such a simple strategy has been shown to achieve a regret of $\tilde{\mathcal{O}}(\sqrt{T} + T\sqrt{A/M})$ (Lu & Van Roy, 2017; Qin et al., 2022) for $T$ steps in Gaussian linear bandits, where $M$ is the size of the ensemble and $A$ is the number of arms. This regret bound analysis tells us that adding more ensemble members will reduce the regret, although it's not clear how this theoretical result extends to the RL setting.

**Optimistic Ensemble Sampling.** Unfortunately, ensemble sampling may overly explore some unpromising regions, as it treats each member in the ensemble model equally. This may lead to unnecessary or wasteful explorations. To reduce this, we propose an optimistic version of ensemble sampling which we call OPS-MBPO. Specifically, we keep track of the performance of each ensemble member in terms of the accumulated episodic return (alternatively, one could also use the value function for each policy (Agarwal & Zhang, 2022).) We then use this performance to determine the probability of choosing each member, thus gradually discarding unpromising ensemble members. More precisely, at the beginning of $k_{th}$ episode, we sample the policy from the following Boltzmann distribution, instead of uniformly at random,

$$p_k(\pi = \pi_i) := \frac{\exp\left(\sum_{l=1}^{k} R_{\mathcal{E}}(\pi_i, l)/\tau\right)}{\sum_{j=1}^{N \cdot M} \exp\left(\sum_{l=1}^{k} R_{\mathcal{E}}(\pi_j, l)/\tau\right)}, \tag{19}$$

where $\tau$ is a temperature term for controlling the level of optimism, and $R_{\mathcal{E}}(\pi_i, l)$ is the empirical cumulative reward of $\pi_i$ at the $l_{\text{th}}$ episode. When $\tau \to \infty$, we recover uniform ensemble sampling, which we call PS-MBPO.

### 4 EXPERIMENTS

Our empirical evaluation aims to 1) verify the effectiveness of our proposed methods; 2) offer a deeper understanding about the mechanisms that are key to the improved performance; and 3) provide additional ablation studies on non-key components. We start by introducing the experimental setup.

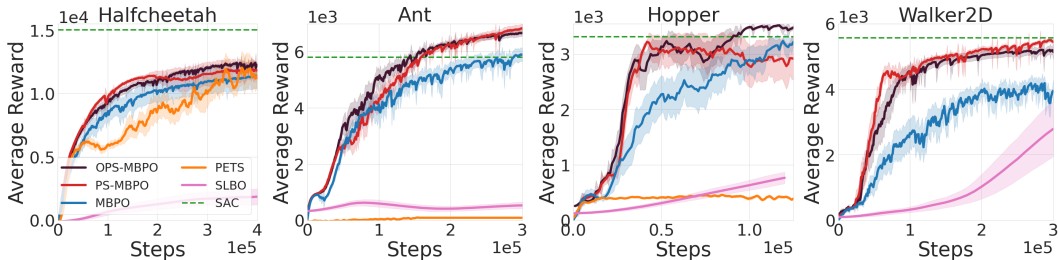

Figure 4: Comparisons on four tasks with dense rewards. The shaded region denotes the one-standard error. The dashed green curve corresponds to the asymptotic performance of SAC at 3M steps. PS-MBPO improves over MBPO across all of the four tasks, and the improvement is more significant on `Ant` and `Walker2D`. OPS-MBPO achieves similar sample efficiency with PS-MBPO.

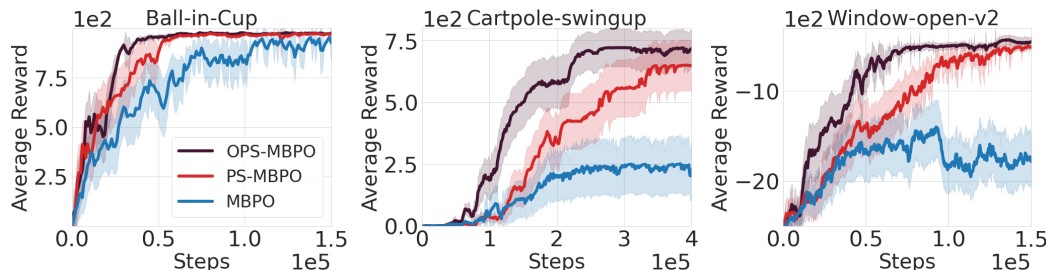

Figure 5: Comparisons on three tasks with sparse rewards. PS-MBPO improves over MBPO, and OPS-MBPO further improves over PS-MBPO in terms of sample efficiency.

## 4.1 EXPERIMENTAL SETUP

We consider seven tasks from OpenAI Gym (Brockman et al., 2016), Deepmind Control Suite (Tunyasuvunakool et al., 2020) and Metaworld (Yu et al., 2020). This includes four dense reward tasks (`ant`, `half-cheetah`, `walker2d` and `hopper`) where the agent receives an immediate reward at each step, and three sparse reward tasks (`ball-in-cup`, `cartpole-swingup` and `window-open-v2`), where the agent receives a reward only if it finishes the corresponding task. For sparse reward tasks, efficiently exploring the environment is more crucial than in dense reward tasks. See Figure 3 for visualizations of the tasks.

For baseline methods, we consider the model-based approaches SLBO (Luo et al., 2019), PETS (Chua et al., 2018) and MBPO (Janner et al., 2019), and a model-free method SAC (Haarnoja et al., 2018). We compare each method in terms of the averaged episode reward, where each episode ends when the time step reaches $1,000$ or the agent reaches the terminal state. To draw more robust conclusions, we repeat each experiment with 10 different random seeds, and report the mean and the standard error. More details can be found in Appendix A and C.

## 4.2 COMPARISON WITH EXISTING METHODS

We report the results on the dense reward tasks in Figure 4. We see that our (O)PS-MBPO outperforms the baselines in all four tasks, including the MBPO method. (We verified that our implementation of MBPO is comparable, or superior, to the performance of the original implementation; see appendix for details.) For example, on `hopper`, our method only requires roughly 40K iterations to reach an average reward around 3,500, whereas MBPO needs around 150K steps.

We report the results on the sparse reward tasks in Figure 5. We first observe that both PS-MBPO and OPS-MBPO outperform MBPO across three tasks, and the improvement is more significant on `Cartpole-swingup` and `Window-open-v2`. In contrast to Figure 4, we also observe that OPS-MBPO further improves PS-MBPO on these three tasks by a significant margin. This confirms the advantage of adopting an optimistic sampling strategy in sparse reward tasks.

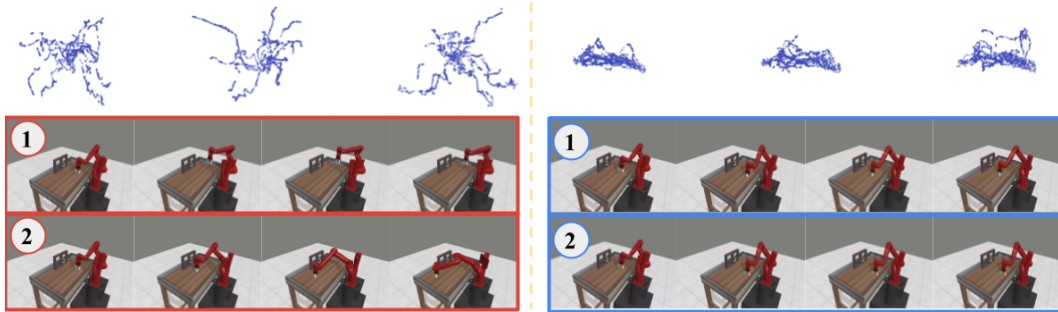

Figure 8: Visualization of the visited state space of PS-MBPO (top left) and MBPO (top right) on `Window-open-v2` at the initial stage of training. We also present two representative trajectories of PS-MBPO (bottom left) and MBPO (bottom right), which are four frames taken from the videos.

## 4.3 ABLATION STUDIES

In this section, we conduct a variety of ablation studies to better understand our proposed method. Additional experiments can be found in the appendix, including ablation study with forced exploration (Phan et al., 2019), random function prior network (Osband et al., 2018) etc.

**Does the gain come from posterior sampling or ensemble?** To assess the importance of posterior sampling, we compare performance where we sample a policy from the posterior at each episode to the standard approach where we use the average policy, computed by averaging the distribution over actions across all ensemble members. The results are presented in Figure 6 with $N = 5$ dynamics networks and $M \in \{1, 3, 5\}$ policy networks. We see that when we disable the sampling procedure, the performance drops significantly. Another interesting observation is that, when *with* posterior sampling, the performance improves as we add more ensemble members (i.e., increase $M$). By contrast, when posterior sampling is *disabled*, increasing $M$ seems doesn't improve the performance. This confirms that posterior sampling is the main factor behind the the improved performance, not just the fact that we have a larger ensemble for both dynamics and policies.

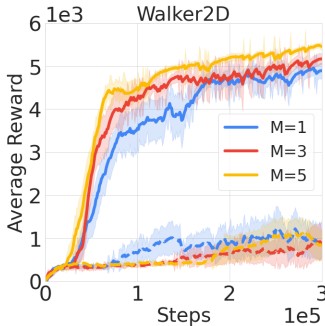

Figure 6: Ablation study on the performance of with (solid curves) and without (dashed curves) the sampling step.

**Effect of $N$ and $M$.** Since we are using the deep ensemble approximation, it is natural to wonder if we could get better performance by using a lager size of ensemble for dynamics models and policies. In Figure 7, we plot the average reward of the last 10 evaluations with 10 different random seeds, using $N$ dynamics models and $M$ policy networks for each dynamics model, by varying $N$ and $M$ in $\{1, 2, 3, 4, 5\}$. We see that increasing both N and M can improve the performance, and that both forms of uncertainty (uncertainty on polices and dynamics) seem to matter.

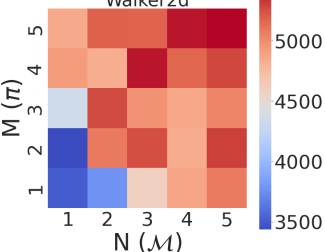

Figure 7: Average reward for varying number of dynamics model ($N$) and policies ($M$).

**Visualization of the state space.** To gain insight into the exploration behavior, we project the high-dimensional states of each trajectories collected by PS-MBPO and MBPO into a two-dimensional space using Umap (McInnes et al., 2018). The visualization of these embeddings can be found in Figure 8. We see that, at the initial phase, PS-MBPO (top left three figures) is more explorative than MBPO (top right three figures), and that this also leads to better final performance (see Figure 5). In addition, we also plot two representative trajectories of PS-MBPO and MBPO for a qualitative comparison in Figure 8 at the same training iterations. Visually, we observe that PS-MBPO can move the robot arm to cover a more diversified regions, whereas MBPO fails to do so and the two trajectories look very similar to each other (this is also reflected in the Umap visualization).

## 5 RELATED WORKS

**Model-based reinforcement learning (MBRL).** Research in MBRL mainly concerns how to learn the dynamics model given the data, and how to use the learned model. The most commonly adopted approach for model learning is using the $L_2$ loss for one-step transitions (Kurutach et al., 2018; Luo et al., 2019; Chua et al., 2018; Janner et al., 2019; Rajeswaran et al., 2020), which is also equivalent to maximum likelihood estimation under a Gaussian assumption. In addition to the one-step training, Hafner et al. (2019); Asadi et al. (2019); Lutter et al. (2021) show that multi-step training can further improves the prediction accuracy for long horizons, but with extra computational costs, which scale quadratically with the number of prediction steps. Another line of work focusses on the objective mismatch problem in MBRL (Ziebart, 2010; Farahmand et al., 2017; Luo et al., 2019; Lambert et al., 2020; Eysenbach et al., 2021), which modify the model training objective to provide performance guarantee for the induced policy in the unknown real environment.

Given the learned model, there are several ways to utilize it. Model predictive control (MPC) (Camacho & Alba, 2013) is a derivative-free optimization method that has been adopted in many prior works (Nagabandi et al., 2018; Chua et al., 2018; Hafner et al., 2019; Fan & Ming, 2021; Lutter et al., 2021). However, MPC is sensitive to the planning horizon and struggles with high-dimensional problems. As a mitigation, Kurutach et al. (2018); Luo et al. (2019); Janner et al. (2019) instead train a policy on top of the model for amortizing the planning cost. Similarly, there are also works that utilize the model to facilitate learning of value functions (Feinberg et al., 2018; Buckman et al., 2018). However, few of these works have investigated the exploration and exploitation tradeoff.

**Exploration and exploitation.** Handling the exploration and exploitation tradeoff is the central problem in online learning. Typical methods can be categorized into the following three classes: 1) *optimism-based* (Jaksch et al., 2010; Pacchiano et al., 2021; Curi et al., 2020); 2) *posterior-sampling-based* (Strens, 2000; Osband et al., 2013; Osband & Van Roy, 2014; Osband et al., 2018; Fan & Ming, 2021); and 3) *information-directed sampling* (Russo & Van Roy, 2014; Nikolov et al., 2019) approaches. Among them, optimism-based methods needs one to construct the confidence set that contains the target model/policy with high probability, which suffers from scalability issues (Osband & Van Roy, 2017); in addition this approach empirically performs worse than Thompson sampling (Chapelle & Li, 2011). Information-directed sampling can be better than optimism-based methods and Thompson sampling, as it directly minimizes the "regret per information bit" (Russo & Van Roy, 2014). However, it relies on estimating the mutual information between random variables, which is especially difficult for high-dimensional continuous random variables (McAllester & Stratos, 2020). Therefore, we focus on posterior sampling. However, different from prior works, we study the effect of approximate inference in an RL setting.

## 6 SUMMARY AND FUTURE WORK

In this paper, we presented PS-MBPO and OPS-MBPO as two algorithms for efficient model-based reinforcement learning in complex environments. We demonstrate that both PS-MBPO and OPS-MBPO can greatly improve the sample efficiency in online reinforcement learning, and surpass various baseline methods by a large margin, especially on sparse reward tasks. We hope that, beyond our specific approach, our analysis can inspire future works to propose improved factorizations of the posterior over policies and models.

In the future, we would like to explore automatically adapting the value of $\lambda$ in an online fashion. (See Section B.2 for some preliminary results.) In addition, we would like to extend the result of Phan et al. (2019) and prove a sublinear regret for PSRL under approximate inference. It would also be interesting to explore epistemic neural networks (Osband et al., 2021) and transformers (Vaswani et al., 2017; Müller et al., 2021) as alternatives to deep ensembles for approximate posterior inference. Lastly, making information directed sampling (Russo & Van Roy, 2014) practical for reinforcement learning problems is another promising direction to further improve the sample efficiency.

**Reproducibility Statement.** We will open source the code for reproducing the results in our paper. For details about our experiments and algorithms, we encourage the reader to checkout Appdendix A for extended backgrounds and Appendix C for the hyperparameters used in our experiments.

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

CONTENTS

## A EXTENDED BACKGROUNDS

### A.1 DYNAMICS MODEL

We use deep ensemble for fitting the environment dynamics. For each network in the ensemble $f_{\boldsymbol{\theta}}$, it takes a whitened state and action pair as input, and predicts the residual of the next state as well as the reward, i.e.,

$$f_{\boldsymbol{\theta}}\left(\frac{\boldsymbol{s}_t - \boldsymbol{\mu}_s}{\boldsymbol{\sigma}_s}, \frac{\boldsymbol{a}_t - \boldsymbol{\mu}_a}{\boldsymbol{\sigma}_a}\right) = \mathcal{N}\left(\begin{bmatrix} \Delta \boldsymbol{s}_t \\ r(\boldsymbol{s}_t, \boldsymbol{a}_t) \end{bmatrix}, \begin{bmatrix} \text{diag}(\boldsymbol{\sigma}_{\Delta \boldsymbol{s}_t}^2), & \mathbf{0} \\ \mathbf{0}, & \sigma_{r_t}^2 \end{bmatrix}\right), \tag{20}$$

where $\boldsymbol{\mu}_s$, $\boldsymbol{\mu}_a$ are the empirical mean of the states and actions, $\boldsymbol{\sigma}_s$ and $\boldsymbol{\sigma}_a$ are the empirical standard deviation of them. Then, the predictions of next state and reward will be

$$\begin{bmatrix} \boldsymbol{s}_{t+1} \\ r_t \end{bmatrix} \sim \mathcal{N}\left(\begin{bmatrix} \boldsymbol{s}_t + \Delta \boldsymbol{s}_t \\ r(\boldsymbol{s}_t, \boldsymbol{a}_t) \end{bmatrix}, \begin{bmatrix} \text{diag}(\boldsymbol{\sigma}_{\Delta \boldsymbol{s}_t}^2), & \mathbf{0} \\ \mathbf{0}, & \sigma_{r_t}^2 \end{bmatrix}\right). \tag{21}$$

Below is our implementation of each individual neural network in JAX (Bradbury et al., 2018).

```python
class GaussianMLP(hk.Module):
  """MLP with Gaussian distribution outputs."""

  def __init__(
      self,
      output_size: int,
      hidden_sizes: Sequence[int],
      *,
      activation=jax.nn.swish,
      min_logvar: float = -10.0,
      max_logvar: float = 2.0,
      name: Optional[str] = None,
  ):
    super().__init__(name=name)
    self.output_size = output_size
    w_init = hk.initializers.VarianceScaling(1.0, 'fan_in', '
                                    truncated_normal')
    self.mlp = hk.nets.MLP(
        hidden_sizes, w_init=w_init, activation=activation,
                                    activate_final=True)
    self.min_logvar = jnp.ones(output_size) * min_logvar
    self.max_logvar = jnp.ones(output_size) * max_logvar
    self.mean_and_logvar = hk.Linear(
        self.output_size * 2, w_init=w_init, name='mean_and_logvar')

  def __call__(self, x):
    h = self.mlp(x)
    mean, logvar = jnp.split(self.mean_and_logvar(h), 2, axis=-1)
    logvar = self.max_logvar - jax.nn.softplus(self.max_logvar - logvar)
    logvar = self.min_logvar + jax.nn.softplus(logvar - self.min_logvar)
    return mean, logvar
```

### A.2 SOFT ACTOR-CRITIC (SAC)

We use SAC for learning the policies. In a highlevel, SAC is a maximum entropy RL algorithm, which typically optimizes the following objective,

$$J(\pi) = \sum_{t=0}^{T} \mathbb{E}_{(\boldsymbol{s}, \boldsymbol{a}) \sim \rho_{\pi}}[r(\boldsymbol{s}, \boldsymbol{a}) + \alpha \mathbb{H}(\pi(\cdot | \boldsymbol{s}))]. \tag{22}$$

As a result, maximum entropy RL algorithm will favor those policies that not only optimize for the reward, but also has a large entropy. This can in turn improve the robustness of the optimized policy.

As for SAC, it searches the policy by iteratively solving the policy evaluation and policy improvement steps.

$$\text{Policy Evaluation: } Q^{\pi_t}(\boldsymbol{s}, \boldsymbol{a}) \leftarrow r(\boldsymbol{s}, \boldsymbol{a}) + \gamma \mathbb{E}_{\boldsymbol{s}' \sim p(\cdot|\boldsymbol{s}, \boldsymbol{a})}[V^{\pi_t}(\boldsymbol{s}')], \tag{23}$$

$$V^{\pi_t}(\boldsymbol{s}) = \mathbb{E}_{\boldsymbol{a} \sim \pi_t(\cdot|\boldsymbol{s})}[Q^{\pi_t}(\boldsymbol{s}, \boldsymbol{a}) - \log \pi_t(\boldsymbol{a}|\boldsymbol{s})]; \tag{24}$$

$$\text{Policy Improvement: } \pi_{t+1} \leftarrow \arg \min_{\pi} \mathbf{d}_{\mathrm{KL}}\left(\pi(\cdot|\boldsymbol{s}_0)| \exp(Q^{\pi_t}(\boldsymbol{s}_0, \cdot))\right). \tag{25}$$

In the practical implementation of SAC, it uses a separate function approximator for the state value to stabilize the training. Specifically, there are three components in SAC, a parameterized state value function $V_{\boldsymbol{\psi}}(\boldsymbol{s})$, a soft Q-function $Q_{\boldsymbol{\theta}}(\boldsymbol{s}, \boldsymbol{a})$, and a policy $\pi_{\boldsymbol{\phi}}(\boldsymbol{a}|\boldsymbol{s})$. The objectives for each component are

$$J_V(\boldsymbol{\psi}) = \mathbb{E}_{\boldsymbol{s} \sim \mathcal{D}}\left[\frac{1}{2}\left(V_{\boldsymbol{\psi}}(\boldsymbol{s}) - \mathbb{E}_{\boldsymbol{a} \sim \pi_{\boldsymbol{\phi}}(\cdot|\boldsymbol{s})}[Q^{\pi_{\boldsymbol{\phi}}}(\boldsymbol{s}, \boldsymbol{a}) - \log \pi_{\boldsymbol{\phi}}(\boldsymbol{a}|\boldsymbol{s})]\right)^2\right], \tag{26}$$

$$J_Q(\boldsymbol{\theta}) = \mathbb{E}_{(\boldsymbol{s}, \boldsymbol{a}) \sim \mathcal{D}}\left[\frac{1}{2}\left(Q_{\boldsymbol{\theta}}(\boldsymbol{s}, \boldsymbol{a}) - \hat{Q}(\boldsymbol{s}, \boldsymbol{a})\right)^2\right], \tag{27}$$

$$J_{\pi}(\boldsymbol{\phi}) = \mathbb{E}_{\boldsymbol{s} \sim \mathcal{D}}\left[\mathbf{d}_{\mathrm{KL}}\left(\pi_{\boldsymbol{\phi}}(\cdot|\boldsymbol{s})| \exp(Q_{\boldsymbol{\theta}}(\boldsymbol{s}, \cdot))\right)\right], \tag{28}$$

where $Z_{\boldsymbol{\theta}}(\cdot)$ is a normalizing constant, and $\hat{Q}(\boldsymbol{s}, \boldsymbol{a})$ is defined as

$$\hat{Q}(\boldsymbol{s}, \boldsymbol{a}) := r(\boldsymbol{s}, \boldsymbol{a}) + \gamma \mathbb{E}_{\boldsymbol{s}' \sim p(\cdot|\boldsymbol{s}, \boldsymbol{a})}\left[V_{\bar{\boldsymbol{\psi}}}(\boldsymbol{s}')\right]. \tag{29}$$

Additionally, $\bar{\boldsymbol{\psi}}$ is the exponentially moving average of the weights of the value network, and $J_{\pi}(\boldsymbol{\phi})$ can be optimized with reparameterization trick under Gaussian case, which can further reduces the variance of the gradient estimator and hence stabilizes the training. We adopt the SAC implementation from Acme (Hoffman et al., 2020).

## A.3 REVISITING MBPO, PETS AND ME-TRPO

Popular model-based reinforcement learning algorithms such as ME-TRPO (Kurutach et al., 2018), PETs (Chua et al., 2018) and MBPO (Janner et al., 2019) typically repeat the following three steps: 1) train a dynamics model (or an ensemble of models) $q(\mathcal{M}|\mathcal{D}_{\mathcal{E}})$; 2) train/extract a policy $\pi^{\star}$ from the learned model; 3) collect data from the environment with the policy. Consequently, their policy is (approximately) equivalent to the one obtained by solving

$$\pi^{\star} = \arg \max_{\pi \in \Pi} \mathbb{E}_{\mathcal{M}}[R_{\mathcal{M}}(\pi)] = \arg \max_{\pi \in \Pi} \int R_{\mathcal{M}}(\pi) q(\mathcal{M}|\mathcal{D}_{\mathcal{E}}) d\mathcal{M}, \tag{30}$$

where the posterior of the model $\mathcal{M}$ is approximated by an ensemble of neural networks, $\Pi$ is the search space of policies, and the cumulative reward $R_{\mathcal{M}}(\pi)$ for an episode of length $H$ of a policy $\pi$ under dynamics model $\mathcal{M}$ is defined as

$$R_{\mathcal{M}}(\pi) = \mathbb{E}\left[\sum_{t=1}^{H} r_{\mathcal{M}}(\boldsymbol{s}_t, \boldsymbol{a}_t)\right] \quad \text{where} \quad \boldsymbol{s}_{t+1} \sim p_{\mathcal{M}}(\boldsymbol{s}|\boldsymbol{s}_t, \boldsymbol{a}_t) \text{ and } \boldsymbol{a}_t \sim \pi(\boldsymbol{a}|\boldsymbol{s}_t). \tag{31}$$

However, the above strategy only accounts for exploitation, so will lead to low data efficiency.

## A.4 POSTERIOR SAMPLING REINFORCEMENT LEARNING

The idea of PSRL is introduced by Strens (2000). The first regret bound $\tilde{\mathcal{O}}(HS\sqrt{AT})$ for PSRL is proved by Osband et al. (2013) for a tabular case with $S$, $A$, $T$, $H$ denotes the number of state, number of actions, number of time steps, and the length of each episode, respectively. In Osband & Van Roy (2017), the bound was improved to $\tilde{\mathcal{O}}(H\sqrt{SAT})$. For the continuous case, Osband & Van Roy (2014) provides the first regret bound $\tilde{\mathcal{O}}(\sqrt{d_K d_E T})$ based on the eluder dimension $d_E$ and Kolmogorov dimension $d_K$. More recently, Fan & Ming (2021) study the regret bound for PSRL under Gaussian process assumption, and obtain a regret bound of $\tilde{\mathcal{O}}(H^{3/2}d\sqrt{T})$. In addition to the bound on Bayesian regret, there is also a line of works studying the worst-case or frequentist regret bound for PSRL (Agrawal & Jia, 2017; Tiapkin et al., 2022b;a), and achieve a regret bound of order $\tilde{\mathcal{O}}(\sqrt{T})$. Nevertheless, all these regret bounds are derived under exact Thompson sampling.

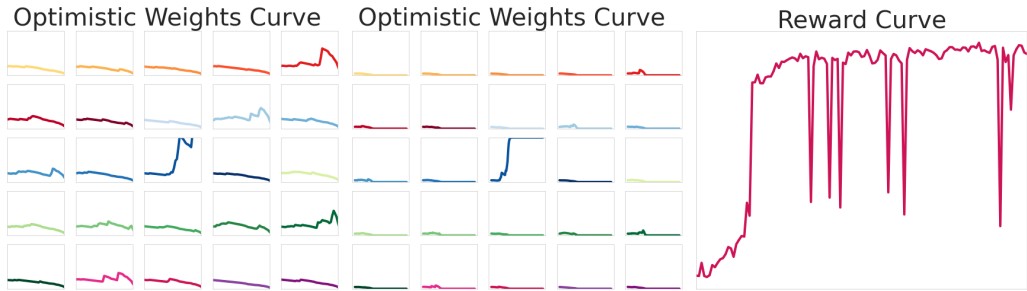

Figure 9: Visualization of the optimistic weights of the first 40K iterations (left) and during the entire training process (middle), and the reward curve (right) on `Hopper`. Each chart in the left and middle figures corresponds to the weights of each single policy.

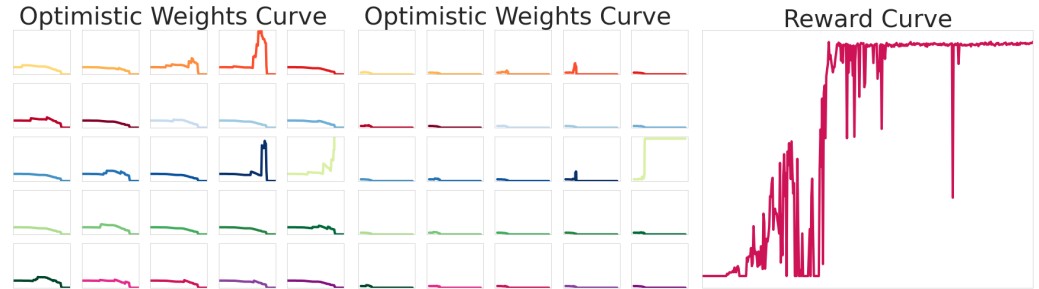

Figure 10: Visualization of the optimistic weights of the first 100K iterations (left) and during the entire training process (middle), and the reward curve (right) on `Cartpole-Swingup`. Each chart in the left and middle figures corresponds to the weights of each single policy.

## B    ADDITIONAL RESULTS FOR (O)PS-MBPO

### B.1    VISUALIZATION OF THE OPTIMISTIC WEIGHTS

For OPS-MBPO, we will maintain the weights for each policy throughout the entire process of online learning. To investigate how those weights evolve, we plot the weights of each policy in Figure 9 and Figure 10, which covers one dense reward task and one sparse reward task. The leftmost figure corresponds to the weights in the initial phase, which will change more rapidly than the later phases. We observe that the weights of some polices will first go up and than go down, and finally it will converge to a single policy. More interestingly, the reward curve in the rightmost figure is also consistent with the pattern in the optimistic weights curve.

### B.2    HOW DOES THE TEMPERATURE TERM $\tau$ AFFECT THE PERFORMANCE?

We further study how does the temperature term will affect the performance on both dense reward and sparse reward tasks. The results can be found in Figure 11 and Figure 12. We observe that the temperature term will affect the convergence speed of the reward on most of the tasks. For some of the tasks, such as `Ant`, `Hopper` and `Walker2d`, it will also affect the converged reward slightly. In general, we recommend the temperature term to be around five times of the best averaged episodic reward that can be achieved.

### B.3    HOW DOES THE SCHEDULE OF $\lambda$ AFFECT THE PERFORMANCE?

Since $\lambda$ plays a role in balancing the effect of approximate inference error and data efficiency, we are interested in how different schedules of $\lambda$ will affect the reward curve. We consider two schemes for adjusting $\lambda$, i.e., 1) a constant schedule; and 2) a linear schedule. For constant schedule, we fix the value of $\lambda$ throughout training, whereas for linear schedule, we decrease the value of $\lambda$ from 1 to 0 linearly. The rightmost figure in Figure 13 visualizes the difference between these two schedules. To make them comparable, we ensure that the areas under both curves are of the same size, so that the

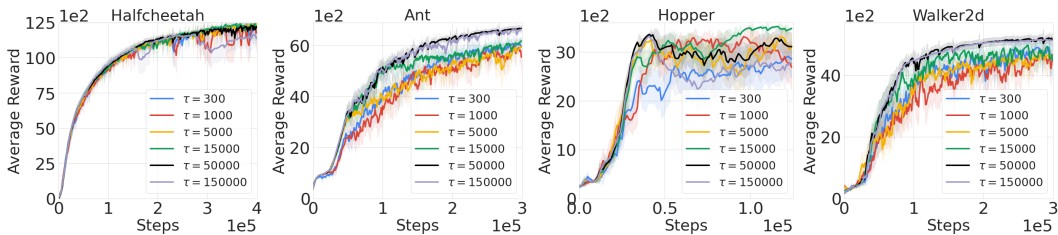

Figure 11: Ablation study on how the choice of the temperature will affect the performance on dense reward tasks. All the experimental setups are the same as those experiments in our main paper, except for the temperature term.

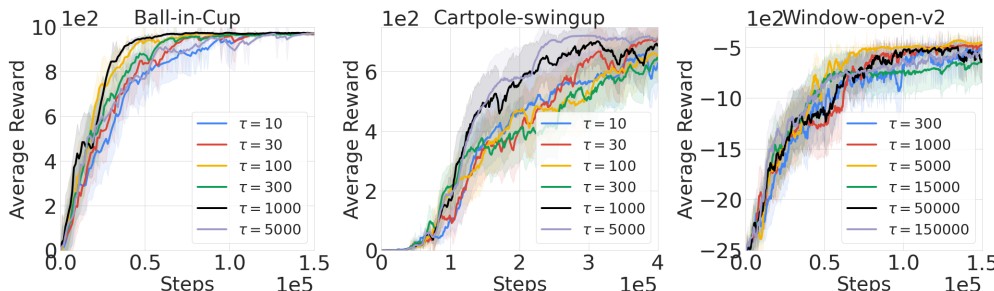

Figure 12: Ablation study on how the choice of the temperature will affect the performance on sparse reward tasks. All the experimental setups are the same as those experiments in our main paper, except for the temperature term.

total amount of real-world data is the same. The comparison on three tasks are shown in the left three figures of Figure 13. We observe that the linear schedule has very little effect on the dense reward tasks, though slightly improves the the final reward in Hopper. For Cartpole-swingup, the performance of linear schedule improves faster than constant schedule, but achieves similar rewards in the end. Nevertheless, we believe that there might be more sophisticated schedules for $\lambda$ that can achieve better performance than the constant schedule, e.g., adapting the value of $\lambda$ based on the model's validation loss. For simplicity, we recommend to use a constant schedule in practice. For sparse reward tasks, the search range of $\lambda$ can be $\{0, 0.1, 0.3, 0.5, 0.7\}$, and $\{0, 0.05, 0.15, 0.3\}$ for dense reward tasks.

In addition to the grid search, we believe it's also possible to adapt the value of $\lambda$ online. We can cast the the problem of choosing the optimal value of $\lambda$ as a bandit problem. The high-level idea of the algorithm is: 1) Initialize a set of possible values for $\lambda$, and treat each value of $\lambda$ as an arm in bandit. 2) Apply any no-regret learning algorithms for solving it, e.g., explore-then-commit. However, we haven't test this algorithm yet, and it would be interesting as a future extension.

## C ADDITIONAL DETAILS ABOUT THE ALGORITHM AND EXPERIMENTS

---

**Algorithm 2** PS-MBPO (abstract formulation)

---

**Require:** Prior distributions $q(\mathcal{M})$, $q(\pi)$ and tuning hyperparameter $\lambda$.
**Require:** Initialize an empty dataset $\mathcal{D}_\mathcal{E}$ for storing data collected from the environment.
 1: **for** $K$ episodes **do**
 2:      Fit the posterior of the policy $q(\pi|\mathcal{D}_\mathcal{E}, \lambda)$ on data $\mathcal{D}_\mathcal{E}$ using equation 13.
 3:      Sample a policy $\pi^k \sim q(\pi|\mathcal{D}_\mathcal{E}, \lambda)$ from the posterior distribution.
 4:      **for** $H$ steps **do**
 5:          Run the policy $\pi^k$ in the environment and add the collected data to $\mathcal{D}_\mathcal{E}$.
 6:      **end for**
 7: **end for**

---

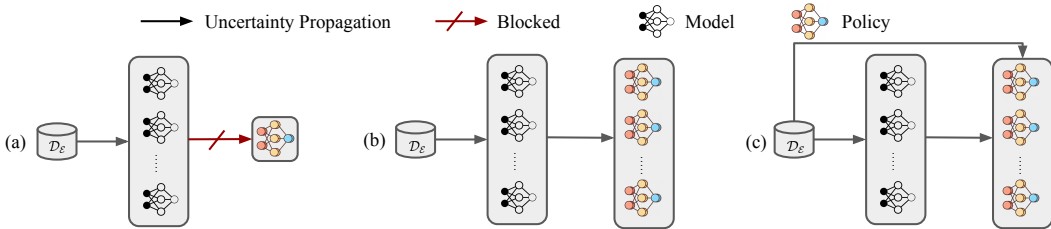

Figure 13: Ablation study on how the schedule of $\lambda$ affect the performance with PS-MBPO. All the experimental setups are the same as the experiments in main paper.

Figure 14: An illustration of the differences between (a) MBPO, (b) PS-MBPO with $\lambda = 0$ and (c) PS-MBPO with $\lambda \in (0, 1)$. MBPO adopt a point estimation to the policy, which is obtained by MAP inference. Thus, the uncertainty propagation from the dynamics model to the policy is blocked. For (b), it implicit assumes the dynamics model captures all the relevant properties of the data. For (c), we add a short cut from data directly to the policy, which is controlled by the value of $\lambda$. Hence, the policy can further utilize the information in the data that are not captured in the dynamics model.

## C.1 ALGORITHM DETAILS

In Algorithm 1, we approximate the posterior of MDPs and policies, i.e., $q(\mathcal{M}|\mathcal{D}_\mathcal{E})$ and $q(\pi|\mathcal{M}, \mathcal{D}_\mathcal{E}, \lambda)$ using deep ensemble, which can be regarded as a finite particle approximation to the posterior. Specifically, $q(\mathcal{M}|\mathcal{D}_\mathcal{E})$ is approximated by $\{\mathcal{M}_{\hat{\boldsymbol{\theta}}_n}\}_{n=1}^{N}$ and $q(\pi|\mathcal{M}_{\hat{\boldsymbol{\theta}}_n}, \mathcal{D}_\mathcal{E}, \lambda)$ is approximated by $\{\pi_{\hat{\boldsymbol{\phi}}_{n,m}}\}_{m=1}^{M}$ for all $n \in [N]$, where both $\mathcal{M}_{\hat{\boldsymbol{\theta}}_n}$ and $\pi_{\hat{\boldsymbol{\phi}}_{n,m}}$ are implemented using a multi-layer perceptron (MLP) with parameters $\hat{\boldsymbol{\theta}}_n$ trained on $\mathcal{D}_\mathcal{E}$ and $\hat{\boldsymbol{\phi}}_{n,m}$ trained on the mixed dataset $\lambda \mathcal{D}_\mathcal{E} + (1 - \lambda)\mathcal{D}_{\mathcal{M}}^{n,m}$, respectively. By mixed dataset $\lambda \mathcal{D}_\mathcal{E} + (1 - \lambda)\mathcal{D}_{\mathcal{M}}^{n,m}$, we mean that for each data point in the training batch, it is with probability of $\lambda$ being sampled from the real data $\mathcal{D}_\mathcal{E}$ and probability of $1 - \lambda$ from the fictitious data $\mathcal{D}_{\mathcal{M}}^{n,m}$.

## C.2 IMPLEMENTATION DETAILS

In this section, we provide the additional details about our algorithm and experiments. We provide a detailed description of our approach in Algorithm 1 and a visual illustration about its difference with MBPO in Figure 14. In terms of the hyperparameters, our choice of them are mostly the same as the ones adopted in MBPO (Janner et al., 2019) and Pineda et al. (2021) for Ant, Halfcheetah, Hopper, Walker2D and Cartpole-swingup, and Eysenbach et al. (2021) for Window-open-v2, which are sufficiently optimized by the authors for MBPO. Specifically, the hyperparameters of MBPO are directly adopted from https://github.com/facebookresearch/mbrl-lib for dense reward tasks. For sparse reward tasks, the hyperparameters are adopted from Eysenbach et al. (2021). The hyperparameters for our method on each task are reported in Table 1. We will also release our code for reproducing all the experiments. Next, we introduce the details about each tasks.

## C.3 TASK DETAILS

**Ant, Halfcheetah, Hopper and Walker2D.** These tasks are taken from the official Github repository of MBPO (Janner et al., 2019), https://github.com/jannerm/mbpo.

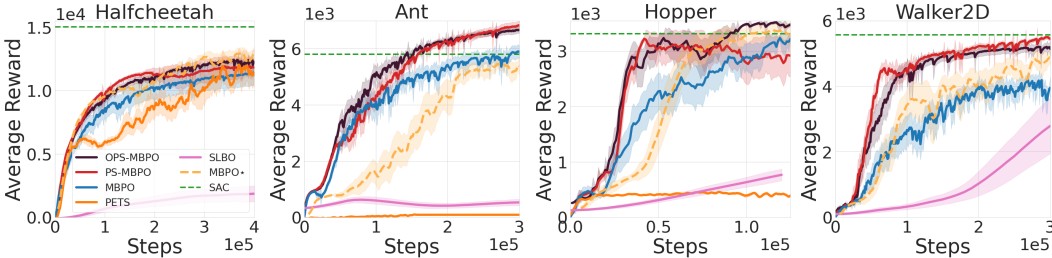

Figure 15: Comparisons on four tasks with dense rewards including `Halfcheetah`, `Ant`, `Hopper` and `Walker2D`. MBPO$^\star$ is the curve from the original paper by Janner et al. (2019). To be noted, in the original implementation of MBPO, they use 7 networks for the ensemble of dynamics model, whereas our implementation only uses 5 networks. But still, our implementation mostly reproduces their results and sometimes is even better.

**Ball-in-Cup and Cartpole-swingup.** These tasks are taken from the deepmind control suite (Tunyasuvunakool et al., 2020). More details can be found in the Github repository at `https://github.com/deepmind/dm_control/`.

**Window-open-v2.** This task is based on the original `Window-open-v2` in Metaworld benchmakr (Yu et al., 2020). The sparse reward is 0 only if the window is within 3 units of the open position, and $-10$ for all other positions.

Table 1: Hyperparameters for each task. $x \to y$ over episodes $a \to b$ denotes a segment linear function, $f(k) = \lfloor \min(\max(x + (k - a)/(b - a) \cdot (y - x), x), y) \rfloor$. We use `Ball`, `Cart`, `Cheetah`, `Walker` and `Window` as abbreviations for `Ball-in-Cup`, `Cartpole-Swingup`, `Halfcheetah`, `Walker2D` and `Window-open-v2` so as to make the table fit in the page.

| Hyper-parameter | Ant | Ball | Cart | Cheetah | Hopper | Walker | Window |
|---|---|---|---|---|---|---|---|
| Replay buffer capacity | $10^6$ | $10^6$ | $10^6$ | $10^6$ | $10^6$ | $10^6$ | $10^6$ |
| Episode length | 1000 | 1000 | 1000 | 1000 | 1000 | 1000 | 250 |
| Number of episodes | 300 | 150 | 400 | 300 | 125 | 300 | 600 |
| Batch size for model | 256 | 256 | 256 | 256 | 256 | 256 | 256 |
| Model update frequency | 250 | 250 | 250 | 250 | 250 | 250 | 250 |
| Model Hidden dim. | 200 | 200 | 200 | 200 | 200 | 200 | 200 |
| Model #Hidden layers | 4 | 4 | 4 | 4 | 4 | 4 | 4 |
| Learning rate for model | $10^{-3}$ | $10^{-3}$ | $10^{-3}$ | $10^{-3}$ | $10^{-3}$ | $10^{-3}$ | $10^{-3}$ |
| Weight decay for model | $10^{-4}$ | $5 \times 10^{-5}$ | $5 \times 10^{-5}$ | $10^{-4}$ | $10^{-4}$ | $10^{-4}$ | $10^{-4}$ |
| #Model ensemble (N) | 5 | 5 | 5 | 5 | 5 | 5 | 5 |
| Validation ratio | 20% | 20% | 20% | 20% | 20% | 20% | 20% |
| Rollout batch size | $10^5$ | $10^5$ | $10^5$ | $10^5$ | $10^5$ | $10^5$ | $5 \times 10^4$ |
| Model horizon | $\begin{array}{c}1 \to 25\\ \text{over episodes}\\ 20 \to 100\end{array}$ | 1 | 1 | 1 | $\begin{array}{c}1 \to 15\\ \text{over episodes}\\ 20 \to 100\end{array}$ | 1 | 1 |
| Batch size for policy | 256 | 256 | 256 | 256 | 256 | 256 | 256 |
| Learning rate for policy | 3e-4 | 3e-4 | 3e-4 | 3e-4 | 3e-4 | 3e-4 | 3e-4 |
| #Policy per model (M) | 5 | 5 | 5 | 5 | 5 | 5 | 5 |
| Discount ($\gamma$) | 0.99 | 0.99 | 0.99 | 0.99 | 0.99 | 0.99 | 0.99 |
| Target entropy | $-4$ | $-0.05$ | $-0.05$ | $-3$ | $-1$ | $-3$ | $-1$ |
| Policy update frequency | 1 | 1 | 1 | 1 | 1 | 1 | 1 |
| Optimizer | Adam | Adam | Adam | Adam | Adam | Adam | Adam |
| $\lambda$ | 0.05 | 0.5 | 0.5 | 0.05 | 0.05 | 0.05 | 0.5 |
| $\tau$ for OPS | 50000 | 1000 | 5000 | 15000 | 15000 | 50000 | 5000 |

# D FORCED EXPLORATION

Forced exploration is proposed in Phan et al. (2019) to improve approximate Thompson sampling for bandit problems. Without properly dealing with the approximate inference error, there will be an extra term in the regret that is linear in $T$, regardless how small the error is. In their paper, they use

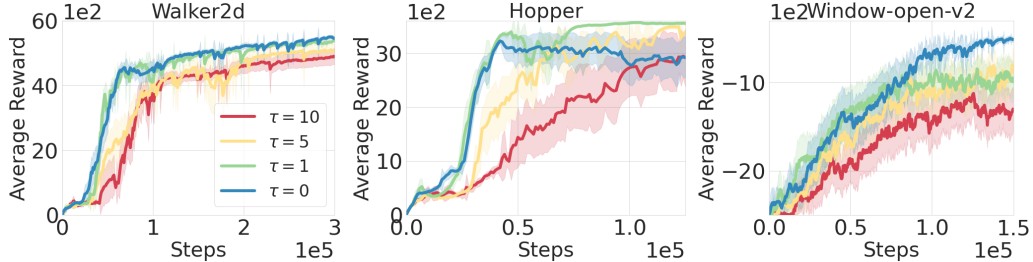

Figure 16: Experiments of forced exploration on `Walker2d`, `Hopper` and `Window-open-v2`. The shaded region denotes the one-standard error. $\tau = 0$ is the one without forced exploration.

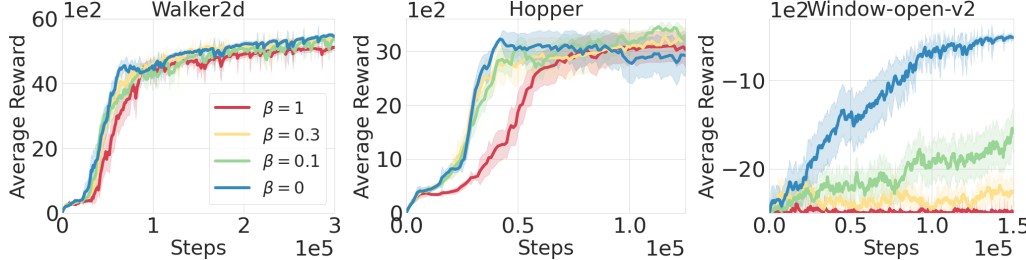

Figure 17: Experiments with random function prior networks (RFPs) on `Walker2d`, `Hopper` and `Window-open-v2`. The shaded region denotes the one-standard error. $\beta = 0$ corresponds to the one without using random function prior networks.

the $\alpha$-divergence for measuring the approximate inference error, defined as

$$D_\alpha(P, Q) = \frac{1 - \int p(x)^\alpha q(x)^{1-\alpha} dx}{\alpha(1 - \alpha)}. \tag{32}$$

The $\alpha$-divergence can capture many divergences, including forward KL divergence ($\alpha \to 1$), backward KL divergence ($\alpha \to 0$), Hellinger distance ($\alpha = 0.5$) and $\chi^2$ divergence ($\alpha = 2$). Different inference methods will give error guarantee measured by $\alpha$-divergence with different $\alpha$.

We are interested in the error guarantee under the reverse KL-divergence, i.e., $\alpha = 0$, as the ensemble sampling (Lu & Van Roy, 2017) provides error guarantees under the reverse Kl-divergence. In Phan et al. (2019), they prove that forced exploration can make the posterior concentrate and hence restore the sub-linear regret bound, if the inference error is bounded by $\alpha$-divergence with $\alpha \le 0$. The reverse KL-divergence falls in this case. Specifically, the forced exploration is a simple method, that maintains a probability of random exploration. This probability decays as $t$, the online steps, grows.

Though the above results only hold for bandit setting and it's unclear for reinforcement learning, we are interested in testing its empirical performance in RL. In our experiments, we consider the following exploration rate

$$p_k(\text{random explore=True}) = \text{Bern}(\tau/k), \tag{33}$$

where $k$ is the index for the episode, and $\tau$ is the hyperparameter for controlling the frequency of forced exploration. As $k$ increases, the random exploration probability will decrease. In our experiments, we consider $\tau \in \{1, 5, 10\}$. All the other settings are the same as our experiments in the main paper. The results are presented in Figure 16. We observe that forced exploration is mostly not helpful in our experiments, except for the `Hopper` task. Moreover, increasing $\tau$ usually make the performance even worse. On the other hand, this may not be so surprising as the forced exploration is designed for approximate Thompson sampling in the bandit setting, and the result may not necessary generalize to the RL setting. We leave the theoretical analysis as a future work.

## E   RANDOM FUNCTION PRIOR

The random function prior (RFP) is proposed in Osband et al. (2018) for improving the uncertainty estimation. The prior network are chosen for modelling the uncertainty that does not come from the

observed data. The RFPs can also be viewed as a regularization in the output space. In contrast to weight space regularization, RFP makes it easier to incorporate different property (e.g., periodicity) of the function to be learned as a prior information. More importantly, when using deep ensemble, incorporating the RFP is fairly simple. It modifies the original training objective $\ell(f_{\boldsymbol{\theta}}, \mathcal{D})$ by adding an extra regularization term,

$$\ell^{RFP}(f_{\boldsymbol{\theta}}, \mathcal{D}) := \ell(f_{\boldsymbol{\theta}} + \beta f_{\boldsymbol{\theta}_0}, \mathcal{D}), \tag{34}$$

where $\beta$ is a scaling term for adjusting the effect of the prior, $f_{\boldsymbol{\theta}_0}$ is the prior network which is held fixed during training. Hence, we also conduct experiments with RFPs in our experiments to investigate how does the RFPs will affect the learning of dynamics models.

We vary the value of $\beta$ in $\{0.1, 0.3, 1\}$. The results are reported in Figure 17. Firstly, by properly choosing the value of $\beta$, RFPs slightly improve the performance on `Hopper`, and don't affect the performance a lot on `Walker2D`. However, for `Window-open-v2`, RFPs will hurt the performance a lot. We conjecture that this might because our choice of the prior function on `Window-open-v2` is not suitable for the task, i.e., the reward is sparse in `Window-open-v2`, but the RFPs don't induce sparsity on the predictions.

Nevertheless, one interesting observation is that both forced exploration and RFPs seem to help on `Hopper`, and their overall pattern on three tasks is a bit consistent. Therefore, it would be interesting to figure out if there is a deep connection between the forced exploration and RFPs.

# F   ADDITIONAL UMAP VISUALIZATIONS

We provide the Umap visualization of the state embeddings of PS-MBPO and MBPO during training in Figure 18 and Figure 19. We observe that PS-MBPO will mostly cover a more broad range of the embedding space, and its pattern also evolves more rapidly than MBPO.

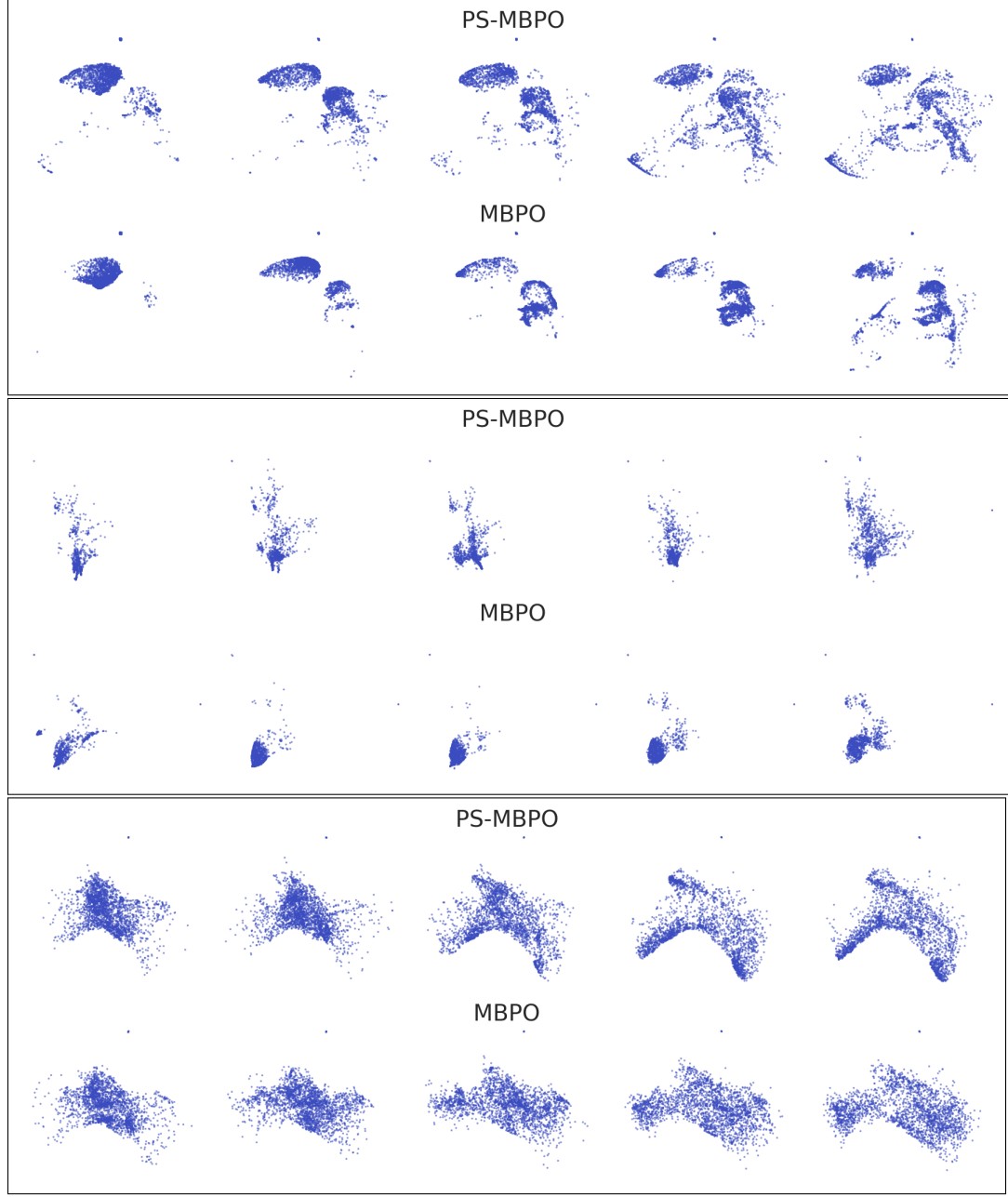

Figure 18: Visualization of the Umap embeddings of PS-MBPO and MBPO from consecutive training periods on `Hopper`, `Ant` and `Halfcheetah` (from top to bottom).

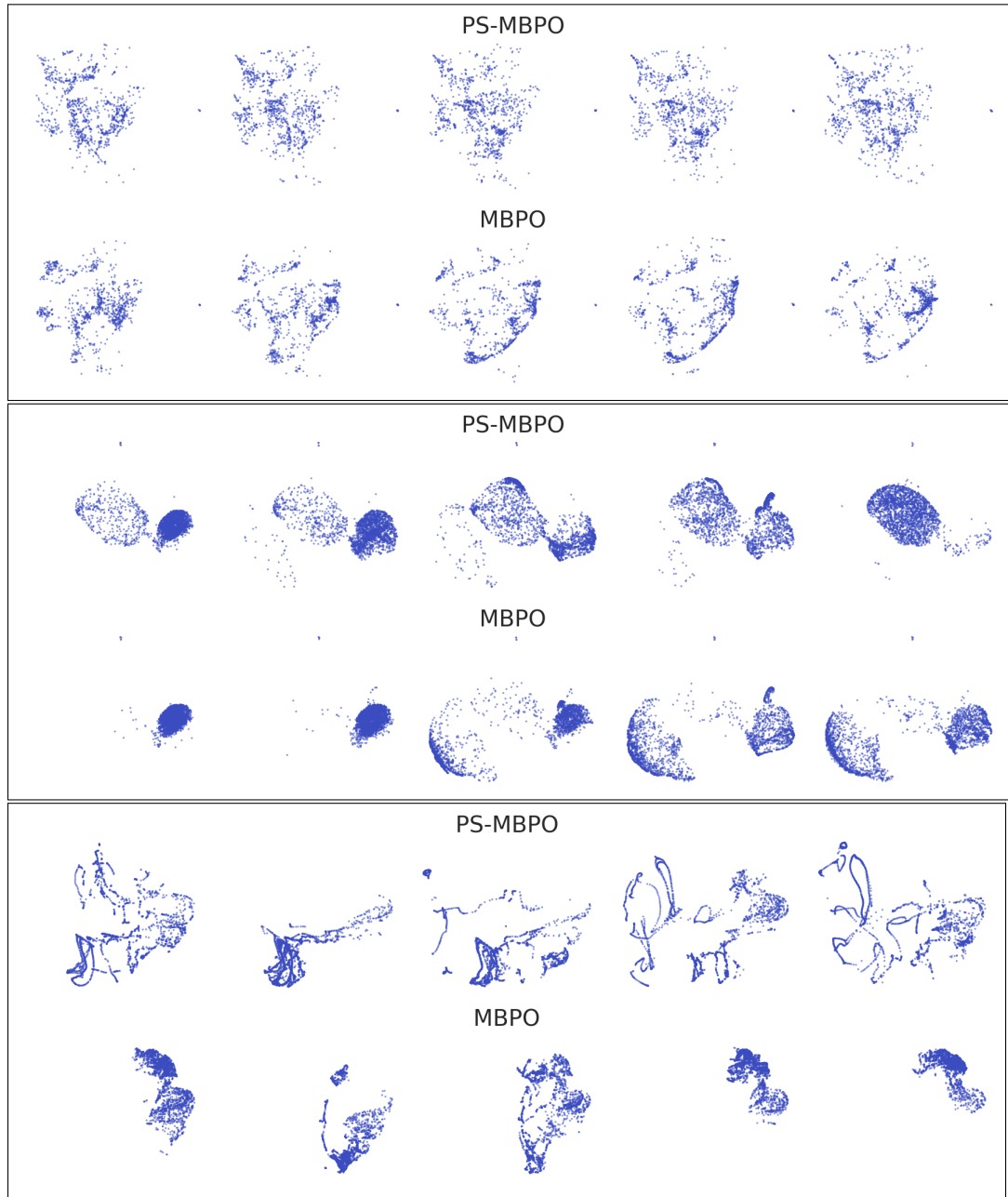

Figure 19: Visualization of the Umap embeddings of PS-MBPO and MBPO from consecutive training periods on `Walker2d`, `Ball-in-Cup` and `Cartpole-swingup` (from top to bottom).

## G PROOFS

In this section, we present the proof of Theorem 1. The proof of this theorem is inspired by the techniques in Russo & Van Roy (2016), with some additional modifications to extend the results from bandit setting to the reinforcement learning setting.

### G.1 PROOF OF THEOREM 1

**Theorem 1** *For $K$ episodes, the Bayesian regret of posterior sampling reinforcement learning algorithm $\mathscr{A}$ with any approximate posterior distribution $q_k$ at episode $k$ is upper bounded by*

$$\sqrt{CK(HR_{\max})^2\mathbb{H}(\pi^\star)} + 2HR_{\max}\sum_{k=1}^{K}\sqrt{\mathbb{E}\left[\mathbf{d}_{KL}\left(q_k(\pi)\mid p_k(\pi)\right)\right]}, \tag{7}$$

*where $\mathbb{H}(\pi^\star)$ is the entropy of the prior distribution of optimal polices, i.e., $p(\pi) = \int \delta(\pi|\mathcal{M})p(\mathcal{M})d\mathcal{M}$, $C$ is some problem-dependent constant and $\mathbf{d}_{KL}\left(\cdot\mid\cdot\right)$ is the KL-divergence.*

*Proof:* Recall the definition of Bayesian regret,

$$\text{BayesianRegret}(T, \mathscr{A}, p(\mathcal{M})) := \mathbb{E}\left[\text{Regret}(T, \mathscr{A}, M)\right] = \mathbb{E}\left[\sum_{k=1}^{K}\Delta_k\right]. \tag{35}$$

Let's denote history at the beginning of episode $k$ as $H_k$. Then, we can rewrite the Bayesian regret as

$$\text{BayesianRegret}(T, \mathscr{A}, p(\mathcal{M})) = \sum_{k=1}^{K}\mathbb{E}_{H_k}\left[\mathbb{E}\left[\Delta_k|H_k\right]\right]. \tag{36}$$

By doing so, we can bound each term $\mathbb{E}[\Delta_k|H_k]$ separately. For convenience, we define $\mathbb{E}_k[\Delta_k] := \mathbb{E}[\Delta_k|H_k]$. Then, by Lemma 1, we can further decompose it into,

$$\mathbb{E}[\Delta_k|H_k] = G_k + D_k, \tag{37}$$

where

$$G_k := \int \sqrt{q_k(\pi)p_k(\pi)}\left(\mathbb{E}_k\left[V_{\pi,1}^M(\boldsymbol{s}_1)|\pi^\star = \pi\right] - \mathbb{E}_k\left[V_{\pi,1}^M(\boldsymbol{s}_1)\right]\right)d\pi \tag{38}$$

and

$$D_k := \int \left(\sqrt{p_k(\pi)} - \sqrt{q_k(\pi)}\right)\left(\sqrt{p_k(\pi)}\mathbb{E}_k\left[V_{\pi,1}^M(\boldsymbol{s}_1)|\pi^\star = \pi\right] + \sqrt{q_k(\pi)}\mathbb{E}_k\left[V_{\pi,1}^M(\boldsymbol{s}_1)\right]\right)d\pi. \tag{39}$$

Then, it remains to bound $\sum_{k=1}^{K}\mathbb{E}[G_k]$ and $\sum_{k=1}^{K}\mathbb{E}[D_k]$. By Lemma 2, we can bound the sum of expectation of $D_k$ by

$$\sum_{k=1}^{K}\mathbb{E}[D_k] \le 2HR_{\max}\sum_{k=1}^{K}\sqrt{\mathbb{E}\left[\mathbf{d}_{\text{KL}}\left(q_k\mid p_k\right)\right]}. \tag{40}$$

By Lemma 3, the upper bound for the sum of the expectation of $G_k$ is

$$\sum_{k=1}^{K}\mathbb{E}[G_k] \le \sqrt{CK\left((HR_{\max})^2/2\right)\mathbb{H}(\pi^\star)}. \tag{41}$$

Hence, the term $D_k$ captures the regret incurred by the approximate inference error, and $G_k$ captures the standard regret for Thompson sampling, which is of order $\tilde{\mathcal{O}}(\sqrt{K})$. By combining them together, we finally arrive at the upper bound of the Bayesian regret

$$\begin{aligned}\text{BayesianRegret}(T, \mathscr{A}, p(\mathcal{M})) \le &\sqrt{CK(HR_{\max})^2\mathbb{H}(\pi^\star)}\\ &+ 2HR_{\max}\sum_{k=1}^{K}\sqrt{\mathbb{E}\left[\mathbf{d}_{\text{KL}}\left(q_k(\pi)\mid p_k(\pi)\right)\right]}.\end{aligned} \tag{42}$$

$\square$

### G.2 Supporting Lemmas

**Lemma 1** *For each time $k = 1, ..., K$, we have*

$$\mathbb{E}\left[\Delta_k | H_k\right] = \mathbb{E}\left[V_{\pi^\star,1}^M(\boldsymbol{s}_1) - V_{\pi^k,1}^M(\boldsymbol{s}_1) | H_k\right] \coloneqq \mathbb{E}_k\left[V_{\pi^\star,1}^M(\boldsymbol{s}_1) - V_{\pi^k,1}^M(\boldsymbol{s}_1)\right] = G_k + D_k, \quad (43)$$

*where*

$$G_k \coloneqq \int \sqrt{q_k(\pi)p_k(\pi)}\left(\mathbb{E}_k\left[V_{\pi,1}^M(\boldsymbol{s}_1) | \pi^\star = \pi\right] - \mathbb{E}_k\left[V_{\pi,1}^M(\boldsymbol{s}_1)\right]\right) d\pi \quad (44)$$

*and*

$$D_k \coloneqq \int \left(\sqrt{p_k(\pi)} - \sqrt{q_k(\pi)}\right)\left(\sqrt{p_k(\pi)}\mathbb{E}_k\left[V_{\pi,1}^M(\boldsymbol{s}_1) | \pi^\star = \pi\right] + \sqrt{q_k(\pi)}\mathbb{E}_k\left[V_{\pi,1}^M(\boldsymbol{s}_1)\right]\right) d\pi. \quad (45)$$

*Proof:* Conditioning on the history $H_k$, we can write the conditional Bayesian regret as

$$\mathbb{E}_k\left[V_{\pi^\star,1}^M(\boldsymbol{s}_1) - V_{\pi^k,1}^M(\boldsymbol{s}_1)\right] \quad (46)$$

$$= \int p_k(\pi)\mathbb{E}_k\left[V_{\pi,1}^M(\boldsymbol{s}_1) | \pi^\star = \pi\right] d\pi - \int q_k(\pi)\mathbb{E}_k\left[V_{\pi,1}^M(\boldsymbol{s}_1) | \pi^k = \pi\right] d\pi \quad (47)$$

$$= \int p_k(\pi)\mathbb{E}_k\left[V_{\pi,1}^M(\boldsymbol{s}_1) | \pi^\star = \pi\right] d\pi - \int q_k(\pi)\mathbb{E}_k\left[V_{\pi,1}^M(\boldsymbol{s}_1)\right] d\pi \quad (48)$$

$$= G_k + D_k, \quad (49)$$

where the second equality holds because the value function is independent of the instantiation of the policy $\pi^k$ when given the history $H_k$. $\qquad\square$

**Lemma 2** *For any $k = 1, ..., K$, we have*

$$\sum_{k=1}^{K}\mathbb{E}[D_k] \leq 2HR_{\max}\sum_{k=1}^{K}\sqrt{\mathbb{E}\left[\mathbf{d}_{KL}\left(q_k | p_k\right)\right]}. \quad (50)$$

*Proof:* Recall $D_k$,

$$D_k \coloneqq \int \left(\sqrt{p_k(\pi)} - \sqrt{q_k(\pi)}\right)\left(\sqrt{p_k(\pi)}\mathbb{E}_k\left[V_{\pi,1}^M(\boldsymbol{s}_1) | \pi^\star = \pi\right] + \sqrt{q_k(\pi)}\mathbb{E}_k\left[V_{\pi,1}^M(\boldsymbol{s}_1)\right]\right) d\pi \quad (51)$$

By using the Cauchy-Schwarz inequality, we have

$$D_k \leq \left(\sqrt{\int \left(\sqrt{p_k(\pi)} - \sqrt{q_k(\pi)}\right)^2 d\pi}\right)$$
$$\cdot \left(\sqrt{\int p_k(\pi)\mathbb{E}\left[V_{\pi,1}^M(\boldsymbol{s}_1) | \pi^\star = \pi\right]^2 d\pi} + \sqrt{\int q_k(\pi)\mathbb{E}_k\left[V_{\pi,1}^M(\boldsymbol{s}_1)\right]^2 d\pi}\right). \quad (52)$$

By the definition of Hellinger distance $\mathbf{d}_{\mathrm{H}}\left(\cdot | \cdot\right)$ between two random variables, we have

$$D_k \leq \mathbf{d}_{\mathrm{H}}\left(q_k | p_k\right)\left(\sqrt{\int p_k(\pi)\mathbb{E}\left[V_{\pi,1}^M(\boldsymbol{s}_1)^2 | \pi^\star = \pi\right] d\pi} + \sqrt{\int q_k(\pi)\mathbb{E}_k\left[V_{\pi,1}^M(\boldsymbol{s}_1)^2\right] d\pi}\right). \quad (53)$$

Since $[\mathbf{d}_{\mathrm{H}}\left(\cdot | \cdot\right)]^2 \leq \mathbf{d}_{\mathrm{KL}}\left(\cdot | \cdot\right)$ and $V_{\pi,1}^M$ is a bounded random variable with $HR_{\max}$ as its upper bound, we have

$$D_k \leq 2\mathbf{d}_{\mathrm{H}}\left(q_k | p_k\right)HR_{\max} \leq 2\sqrt{\mathbf{d}_{\mathrm{KL}}\left(q_k | p_k\right)}HR_{\max}. \quad (54)$$

Hence, we have

$$\sum_{k=1}^{K}\mathbb{E}[D_k] \leq 2HR_{\max}\sum_{k=1}^{K}\sqrt{\mathbb{E}\left[\mathbf{d}_{\mathrm{KL}}\left(q_k | p_k\right)\right]}. \quad (55)$$

$\qquad\square$

**Lemma 3** *For each $k = 1, ..., K$, we have*

$$\sum_{k=1}^{K} \mathbb{E}[G_k] \leq \sqrt{CK\left((HR_{\max})^2/2\right)\mathbb{H}\left(\pi^\star\right)}. \tag{56}$$

*Proof:* Recall the definition of $G_k$,

$$G_k := \int \sqrt{q_k(\pi)p_k(\pi)}\left(\mathbb{E}_k\left[V_{\pi,1}^M(\boldsymbol{s}_1)|\pi^\star = \pi\right] - \mathbb{E}_k\left[V_{\pi,1}^M(\boldsymbol{s}_1)\right]\right)d\pi. \tag{57}$$

Since $V_{\pi,1}^M$ (here, we drop the dependency on $\boldsymbol{s}_1$ for clearness) is a bounded random variable, and more specifically, it's $((HR_{\max})/2)$-sub-Gaussian. Hence, by Lemma 4, the following holds,

$$\mathbb{E}_k\left[V_{\pi,1}^M|\pi^\star = \pi\right] - \mathbb{E}_k\left[V_{\pi,1}^M\right] \leq \left(\frac{HR_{\max}}{2}\right)\sqrt{2\mathbf{d}_{\mathrm{KL}}\left(p_k(V_{\pi,1}^M|\pi^\star = \pi)\middle|\, p_k(V_{\pi,1}^M)\right)}. \tag{58}$$

This gives us that

$$G_k \leq \int \sqrt{q_k(\pi)p_k(\pi)}\left(\frac{HR_{\max}}{2}\right)\sqrt{2\mathbf{d}_{\mathrm{KL}}\left(p_k(V_{\pi,1}^M|\pi^\star = \pi)\middle|\, p_k(V_{\pi,1}^M)\right)}d\pi. \tag{59}$$

Then, we can further rewrite the KL-divergence using the conditional mutual information $\mathbb{I}_k(\cdot;\cdot)$ (i.e., conditioning on the history $H_k$),

$$\iint q_k(\pi)p_k(\pi')\mathbf{d}_{\mathrm{KL}}\left(p_k(V_{\pi,1}^M|\pi^\star = \pi')\middle|\, p_k(V_{\pi,1}^M)\right)d\pi d\pi' = \int q_k(\pi)\mathbb{I}_k(\pi^\star; V_{\pi,1}^M)d\pi. \tag{60}$$

When conditioning on the history $H_k$, the optimal policy $\pi^\star$ and $M$ is independent of the $\pi^k$, hence we have

$$\int q_k(\pi)\mathbb{I}_k(\pi^\star; V_{\pi,1}^M)d\pi = \int q_k(\pi)\mathbb{I}_k(\pi^\star; V_{\pi^k,1}^M|\pi^k = \pi)d\pi = \mathbb{I}_k(\pi^\star; V_{\pi^k,1}^M|\pi^k). \tag{61}$$

By the fact that $\pi^\star$ is jointly independent of $V_{\pi^k,1}^M$ and $\pi^k$ when conditioning on the history $H_k$, hence we have

$$\mathbb{I}_k(\pi^\star; V_{\pi^k,1}^M|\pi^k) = \mathbb{I}_k(\pi^\star; V_{\pi^k,1}^M|\pi^k) + \mathbb{I}_k(\pi^\star; \pi^k). \tag{62}$$

By the chain rule of mutual information, we finally get

$$\mathbb{I}_k(\pi^\star; V_{\pi^k,1}^M|\pi^k) + \mathbb{I}_k(\pi^\star; \pi^k) = \mathbb{I}_k(\pi^\star; (V_{\pi^k,1}^M, \pi^k)). \tag{63}$$

Now, let's define the following function $g_k$ and $C$,

$$g_k(\pi, \pi') := \sqrt{q_k(\pi)p_k(\pi')}\left(\mathbb{E}_k\left[V_{\pi,1}^M|\pi^\star = \pi'\right] - \mathbb{E}_k\left[V_{\pi,1}^M\right]\right). \tag{64}$$

$$C := \max_{k \in \mathbb{Z}_+} \frac{(\int g_k(\pi, \pi)d\pi)^2}{\iint[g_k(\pi, \pi')]^2 d\pi d\pi'}. \tag{65}$$

Thus, we further have

$$\mathbb{I}_k(\pi^\star; (V_{\pi^k,1}^M, \pi^k)) \geq \frac{2}{(HR_{\max})^2}\iint[g_k(\pi, \pi')]^2 d\pi d\pi'. \tag{66}$$

On the other hand, we can rewrite $G_k$ as

$$G_k = \int g_k(\pi, \pi)d\pi. \tag{67}$$

By rearranging the terms, we get

$$\frac{G_k^2}{\mathbb{I}_k(\pi^\star; (\pi^k, V_{\pi^k,1}^M))} \leq \frac{((HR_{\max})^2/2)\left(\int g_k(\pi, \pi)d\pi\right)^2}{\iint\left[g_k(\pi, \pi')\right]^2 d\pi d\pi'} \leq C\left((HR_{\max})^2/2\right). \tag{68}$$

Hence,

$$G_k \leq \sqrt{C\left((HR_{\max})^2/2\right) \mathbb{I}_k(\pi^\star; (\pi^k, V^M_{\pi^k,1}))}. \tag{69}$$

Hence, we have

$$\sum_{k=1}^K \mathbb{E}[G_k] \leq \sum_{k=1}^K \mathbb{E}\left[\sqrt{C\left((HR_{\max})^2/2\right) \mathbb{I}_k(\pi^\star; (\pi^k, V^M_{\pi^k,1}))}\right] \tag{70}$$

$$= \sqrt{C\left((HR_{\max})^2/2\right)} \sum_{k=1}^K \mathbb{E}\left[\sqrt{\mathbb{I}_k(\pi^\star; (\pi^k, V^M_{\pi^k,1}))}\right] \tag{71}$$

$$\leq \sqrt{C\left((HR_{\max})^2/2\right)} \sqrt{K\mathbb{E}\left[\sum_{k=1}^K \mathbb{I}_k(\pi^\star; (\pi^k, V^M_{\pi^k,1}))\right]} \tag{72}$$

$$= \sqrt{CK\left((HR_{\max})^2/2\right)} \sqrt{\mathbb{E}\left[\sum_{k=1}^K \mathbb{I}_k(\pi^\star; (\pi^k, V^M_{\pi^k,1}))\right]} \tag{73}$$

$$\leq \sqrt{CK\left((HR_{\max})^2/2\right) \mathbb{H}(\pi^\star)}, \tag{74}$$

where the last inequality holds because of the chain rule of mutual information

$$\mathbb{E}\left(\sum_{k=1}^K \mathbb{I}_k(\pi^\star; (\pi^k, V^M_{\pi^k,1}))\right) = \mathbb{E}\left[\sum_{k=1}^K \mathbb{I}(\pi^\star; (\pi^k, V^M_{\pi^k,1})|H_k)\right] \tag{75}$$

$$= \mathbb{E}\left[\mathbb{I}(\pi^\star; (\pi^1, V^M_{\pi^1,1}, \pi^2, V^M_{\pi^2,1}, ..., \pi^K, V^M_{\pi^K,1})\right] \tag{76}$$

$$= \mathbb{H}(\pi^\star) - \mathbb{E}[\mathbb{H}(\pi^\star|(\pi^1, V^M_{\pi^1,1}, \pi^2, V^M_{\pi^2,1}, ..., \pi^K, V^M_{\pi^K,1}))] \tag{77}$$

$$\leq \mathbb{H}(\pi^\star). \tag{78}$$

$\square$

**Remark 1** *When the number of policies $|\Pi|$ is finite, and the value function $V^M_{\pi,1}$ is linear with its parameter lives in $\mathbb{R}^d$, then $C$ can be upper bounded by $d$, i.e., $C \leq d$.*

**Lemma 4 (Russo & Van Roy (2016))** *Suppose that there is a $H_k$-measurable random variable $\eta$, such that for each $\pi \in \Pi$, $V^M_{\pi,1}$ is a $\eta$-sub-Gaussian random variable when conditioned on $H_k$, then for every $\pi, \pi' \in \Pi$, the following holds with probability 1,*

$$\mathbb{E}_k[V^M_{\pi,1}|\pi^\star = \pi'] - \mathbb{E}_k[V^M_{\pi,1}] \leq \eta\sqrt{2\mathbf{d}_{KL}\left(p_k(V^M_{\pi,1}|\pi^\star = \pi') \,\middle|\, p_k(V^M_{\pi,1})\right)}. \tag{79}$$

