# OpenReview forum: "Posterior Sampling Model-based Policy Optimization under Approximate Inference"
_ICLR.cc/2023/Conference — Submitted to ICLR 2023_

### Official Review · Reviewer_E1oc · 2022-10-25

**Confidence:** 4
**Correctness:** 2
**Technical Novelty And Significance:** 2
**Empirical Novelty And Significance:** 2
**Recommendation:** 3

**Clarity, Quality, Novelty And Reproducibility:**

It is not hard to understand the paper. Given the literature on approximate versions of PSRL, I am not sure how novel this paper is. The code is not included in the submission, but the authors say in the paper that they will open source the code.

**Strength And Weaknesses:**

The paper attempts to provide regret guarantees for approximate versions of PSRL, and proposes one simple version motivated by the analysis. To understand the proposed algorithm better, the authors conducted various ablation studies.

There are several weaknesses in the analysis. First I am not sure whether the regret bound in Theorem 1 is correct. For example, why the inequality (74) holds? I guess the chain rule is used here, but I do not see how to use it. Now suppose the regret bound is correct. It is not discussed in the paper how the first term on equation 7 depends on S, A, H. For example, the entropy of the optimal policy could be pretty large; also how the constant C depends the problem parameters? In addition, it seems that the paper does not try to bound the second term on equation 7 for the proposed algorithm. I think the bound on the second term is really needed; otherwise, I could not see how tight the regret bound in terms of the problem parameters.

In terms of experiments, I am not sure why other approximate versions of PSRL are not included as baseline methods, for example, bootstrap DQN [1], RLSVI [2], hypermodels [3], hyperDQN [4]. Although most of them are value-based methods, I think they still need to be included in the experiments for comparison.

[1] Ian Osband, Charles Blundell, Alexander Pritzel, Benjamin Van Roy, Deep Exploration via Bootstrapped DQN, NeurIPS 2016.\
[2] Ian Osband, Benjamin Van Roy, Daniel J. Russo, Zheng Wen, Deep Exploration via Randomized Value Functions, JMLR 2019.\
[3] Vikranth Dwaracherla, Xiuyuan Lu, Morteza Ibrahimi, Ian Osband, Zheng Wen, Benjamin Van Roy, Hypermodels for Exploration, ICLR 2020.\
[4] HyperDQN: A Randomized Exploration Method for Deep Reinforcement Learning, ICLR 2021.


**Summary Of The Paper:**

The paper proposes an approximate version of posterior sampling reinforcement learning (PSRL). Empirical results show that the proposed algorithm outperforms the chosen baseline methods in some benchmark tasks.

**Summary Of The Review:**

As mentioned above, both analysis and experiments need to be improved.

---

### Official Review · Reviewer_eWBV · 2022-10-25

**Confidence:** 3
**Correctness:** 4
**Technical Novelty And Significance:** 4
**Empirical Novelty And Significance:** 4
**Recommendation:** 8

**Clarity, Quality, Novelty And Reproducibility:**

## Clarity
This paper is well-written and is easy to read.

## Quality
The proposed method is inspired from a theoretical consideration, and is of high quality.

## Novelty
As far as I know, the proposed method is novel.

## Reproducibility
Since the authors declare to publish the source code, the reproducibility will be ok. In addition, the supplementary materials are very rich and will enhance reproducibility more.

**Strength And Weaknesses:**

## Strengths
- The proposed method is well-motivated from a theoretical perspective. Although the theoretical finding is simple, it is easy to understand and is actually useful to develop the proposed method.
- The proposed method achieves solid improvement over the baseline methods.
- The paper is well-organized and is easy to read.
- The additional results in Section 4.3 are insightful to understand the behavior of the proposed method.

## Weaknesses
- Optimal value or scheduling of $\lambda$ is not clear, as the authors discuss in the future work section. In real applications where it is costly to deploy an agent to the environment, manual tuning of $\lambda$ should be avoided. Although I don't ask the authors to completely address the issues, I would like to ask the authors to discuss how to roughly decide $\lambda$ without deploying the agent to the environment. Will the quality of the posterior be relevant?

**Summary Of The Paper:**

The present paper is concerned about posterior sampling reinforcement learning (PSRL), which is an extension of Thompson sampling to RL; an MDP is sampled from the posterior distribution of MDP given data, the optimal policy for the sampled MDP is computed and is used to collect more data. The performance of this algorithm is analyzed in terms of Bayesian regret.

The authors are concerned about the case where only an approximate inference is feasible, where an approximate posterior $q$ is substituted for the true posterior. The authors first show that in this case there could be a better alternative than utilizing the optimal policy for the MDP sampled from the approximate posterior in Proposition 1. Motivated by this observation, the authors propose to derive the policy not only from the sampled model but also from the raw data with hyperparameter $\lambda\in[0,1]$ balancing between them. In the proposed algorithm, $\lambda$ is used to construct a mixed dataset, where the original data and the synthetic data are mixed in the ratio of $\lambda:1-\lambda$. The authors provide two approaches to sample policies from the posterior.

The authors empirically evaluate the performance of the proposed method. The result shows that the proposed algorithm achieves better performance than the baseline algorithms, especially for the tasks with sparse rewards. The authors conduct ablation studies to understand the mechanism of the proposed method.

**Summary Of The Review:**

I would like to suggest to accept this paper, given that the proposed method is well motivated and is well investigated empirically. But I would encourage the authors to include more detailed discussion on how to decide $\lambda$ before deployment.

---

### Official Review · Reviewer_m7LJ · 2022-10-27

**Confidence:** 3
**Correctness:** 3
**Technical Novelty And Significance:** 3
**Empirical Novelty And Significance:** 3
**Recommendation:** 6

**Clarity, Quality, Novelty And Reproducibility:**

The intuition of the algorithm is clearly explained, and the results are strongly supported by the experimental performances and ablation analysis. However there are some vagueness brought by misuse of notations. The methodology is simple, novel, and powerful.

**Strength And Weaknesses:**

The intuition is fairly reasonable, introducing a parameter that extends the hypothesis class reduces the loss, so I trust the effectiveness of the design and the superiority of the empirical performance.

However, I think the procedure of the algorithm has not been explained properly, and the notations are disorganized.

Despite the strong empirical justification, I still find the following parts somewhat hindering the understanding of this work. Especially, some notations throughout the methodology and Algorithm 1 is misleading.

- Throughout the paper, the authors are adopting the notation “$q$” to represent both the dynamics models $q(M\mid D_M)$ and the policy networks $q(\mathcal{M}\mid \mathcal{D}_{\mathcal{E}})$, and sometimes $q(\pi \mid \mathcal{M})$. As here $q$ here is not the exact probability and are approximated via different approaches, it seems to me that it is better to use different notations for different approximation networks.
- In (14) and (15), it is clear that $q(\pi \mid \mathcal{M})$ is usually the optimal policy of $\mathcal{M}$ and is estimated using network $q(\theta)$, but what does $q(\pi \mid \mathcal{D}_{\mathcal{E}})$ represent? What is the corresponding procedure in Algorithm 1? (In my understanding $q(\phi)$ is $q(\pi \mid \mathcal{M})$? So what is $q(\pi \mid D)$?)
- In line 11 of Algorithm 1, how to create mixed dataset $\lambda \mathcal{D}_1 + (1 - \lambda) \mathcal{D}_2$? With the presence of $\lambda$, a simple merge does not seem to do the job. Does it mean with probability $\lambda$ sampling from the first and with probability $1 - \lambda$ sampling from the second?
- Notations in (4), when the regret is defined on $M^*$, one cannot take expectation over $p(\mathcal{M})$, I think the notations should be unified (either using $M^*$ or $M$ for both).

I would be willing to raise my score if the algorithm can be clarified.

Minor questions:

- How does the optimal $\lambda$ change across tasks. i.e. how hard is it to tune $\lambda$?

**Summary Of The Paper:**

This paper considers the problem of approximate posterior sampling in model-based policy optimization. The main contribution of this paper is a novel algorithm called PS-MBPO that utilizes the structure within the approximated posterior. By decomposing the posterior into a posterior on the model $\mathcal{M}$ and a delta function indicating the procedure of choosing $\pi$ according to $\mathcal{M}$, the authors proposes to use an alternative function that interpolates between the delta function and the posterior of $\pi$. The proposed algorithm, PS-MBPO, together with its optimistic variant, OPS-MBPO, shows improvements on tasks with both dense rewards and sparse rewards. Moreover, OPS-MBPO outperforms PS-MBPO more significantly in sparse reward tasks.


**Summary Of The Review:**

This paper provides strong empirical validations of the proposed algorithm, but the theoretical part is not written very clearly. Still a solid contribution.

---

### Official Review · Reviewer_PkUu · 2022-10-29

**Confidence:** 3
**Correctness:** 3
**Technical Novelty And Significance:** 2
**Empirical Novelty And Significance:** 3
**Recommendation:** 6

**Clarity, Quality, Novelty And Reproducibility:**

The paper is globally well written, see specific comments for some details. Except the  description (see specific comments) of PS-MBPO and OPS-MBPO that should be detailed. The results seems correct as far as I checked (I dead not read the appendix in details). Adapting PSRL in the approximate inference setting is a valuable contribution. But I still do not get clearly the intuition on why not acting greedily with respect to the sampled MPD from the posterior. And why fundamentally one needs to change the way the sampling policy is selected by factorizing the posterior rather than 'fixing' the approximate posterior. Indeed, in Algorithm 1 one can argue that mxing true transition to the ones generated by the model is just a way to improve the quality of the model and therefore the approximate posterior. That being said it is not clear how shrinking the posterior toward the empirical MDP interact with exploration.

#Specific comments:
- P1, Section 1: We can also see the sampled policy as a sample from the posterior over optimal policy induced by the posterior over MDPs (which is a priori not degenerated).
- P1, Section 1: Precise what the notation \tilde{O} means.
- P1, Section 1: Explain what MBPO means.
- P2, Section 2, (1): Explain what is the expectation E_{\pi,\cM} in particular how the states s_t and actions a_t are sampled.
- P2, Section 2, (2): To be coherent with (1) maybe you should keep the index  {\pi,\cM} for the expectation?
- P3, Section 2, (4):  Can you detail what is p(\cM)?
- P3, Section 2:  "not only attains a [Bayesian] regret". Note that there is also a line of work study the "frequentist" regret of PSRL like algorithms:
   - Shipra Agrawal and Randy Jia. Optimistic posterior sampling for reinforcement learning: worst-case regret bound, 2017.
   - Daniil Tiapkin et al. From Dirichlet to rubin: Optimistic exploration in RL without
 bonuses, 2022.
   - Daniil Tiapkin et al. Optimistic posterior sampling for reinforcement learning with few samples and tight guarantees, 2022.
- P3, Section 2: \delta(\cdot|cM) is a Dirac delta distribution at an optimal policy for the MDP \cM. By the way since there could be several optimal policy you need to precise what you mean by the argmax.  Remove \delta(arg max_\pi R_\cM(\pi)|\cM) = 1 because it is confusing or at least not a correct definition. If you want to detail the density of a distribution you need to specify the measure from which you take the Radon-Nikodym derivative. This remark also holds for Example 1, P4.
For "better empirical performance" you could also refer to Osband and Van Roy, 2017 for the RL setting.
- P3, Section 2, Theorem 1: "prior distribution over [optimal] polices". What do you mean by problem-dependent constant exactly? What is the problem? d_KL is not defined nor the variables from which you take the expectation.
- P3,  Section 2, Proposition 1: What do you mean by optimal q(\pi|\cM)? And it is clear why having (9) is better because we could simultaneously increase the first term in (7). Can you detail why it is not the case?
- P4, Example 1: ON so basically you correct the approximate posterior on the MDP by picking the right distribution q(\pi|\cM). But to do this choice you use information about the true posterior. Then why  not directly use this additional information  to correct  the approximate posterior on the MDP. At least the comparison between the choice q^\delta(\pi|\cM) and the optimized q(\pi|\cM) is not fair since one use more information than the other.
- P4, Section 3.1: What about \lambda \in (0,1) ? And for Figure 2 at this point it is not clear the regret of which algorithm you are talking about.
- P5, Algorithm 1: Equation x.
- P5, Section 3.1: At the end instead of learning a greedy policy with transitions from on model of the ensemble you add true transitions. So you biased you model MDPs from the ensemble toward the empirical MDP and thus concentrate the posterior around the empirical MPD. This make sense if the model MDP are not reliable but intuitively this could be at the price of less exploration since you reduce the variability of the model MDPs (i.e. the posterior could be to concentrated on the empirical model. So I wonder if the problem is just that the learned model are not good enough and you fix it by adding true transitions. Can you elaborate on these points?
- P6, Section 3.2:  It is very hard to follow the description of this section. Maybe you should describe more in details Algorithm 1. What are \theta, \mu,\sigma, x, y ?
- P6, Section 3.3: Why there is no dynamics model in the bandit setting? The dynamic model is the reward function in this case.
- P6, Section 3.3: Can you elaborate on why "ensemble sampling may overly explore some
unpromising regions, as it treats each member in the ensemble model equally" ?
- P7, Section 4.2: I would not say that OPS-MBPO improves PS-MBPO by a significant margin since they both obtain roughly the same asymptotic performance but OPS-MBPO  seems to converge faster.
- P8, Section 4.3, ensemble vs posterior sampling: You average the policies over all N and M?

**Strength And Weaknesses:**

Strengths:
- PSRL sub-optimal under approximate inference.
- New posterior factorization for PSRL under approximate inference.

Weaknesses:
- Description of PS-MBPO and OPS-MBPO.
- Intuition behind the posterior factorization not completely clear.

**Summary Of The Paper:**

The authors study model-based reinforcement learning in episodic Markov decision process (MDP) and in particular the exploration-exploitation dilemma.

Precisely they consider Posterior sampling reinforcement learning (PSRL) under approximate inference. They show that PSRL can be sub-optimal under approximate inference.  To solve this issue they propose an improved factorization of the posterior over the policy where the sampled policy is not anymore an optimal policy of a sampled MDP  from the posterior but a sample from a distribution that depends on the  sampled MDP and the observations collected in the real environment.

They provide a general algorithmic framework under approximate inference and a practical instantiation of it. They also show that empirically the proposed algorithms improve over various baselines on both dense and sparse rewards task from the Deepmind control suite, OpenAI Gym and Metaworld benchmarks.

**Summary Of The Review:**

See above

---

### Decision · Program_Chairs · 2023-01-20

**Decision:**

Reject

**Justification For Why Not Higher Score:**

The theory in the paper does not provide convincing regret bounds for simple special cases. It is also likely that the paper contains technical flaws.

**Justification For Why Not Lower Score:**

N/A

**Metareview: Summary, Strengths And Weaknesses:**

Summary:
This paper studies posterior sampling reinforcement learning (PSRL) under approximate inference. They show that PSRL can be sub-optimal under approximate inference. To solve this issue, they propose an improved factorization of the posterior over the policy where the sampled policy is not an optimal policy of the sampled MDP from the posterior but a sample from a distribution that depends on the sampled MDP and the observations collected in the real environment. They provide a general algorithmic framework under approximate inference and a practical instantiation. They also show that empirically the proposed algorithms improve over various baselines on both dense and sparse rewards tasks from the Deepmind control suite, OpenAI Gym, and Metaworld benchmarks.

Strength:
- Shows that the original PSRL framework is sub-optimal under approximate inference.
- Proposes a new posterior factorization for PSRL under approximate inference.
- The proposed method achieves solid improvement over the baseline methods.

Weakness:
- The biggest concern from the reviewers is the constant in the regret bound. The authors failed to show that C is bounded for even the tabular setting, whereas previous results of PSRL usually have bounds on this.
- The reviewers also cast doubts on the proof of the paper. The metareview cannot verify the correctness of the theoretical results.


Note: missing recent references on online RL.

**Summary Of Ac-Reviewer Meeting:**

The AC has consulted reviewers' opinions about the assumptions of the boundedness of C in the bounds of the paper. The AC agrees that the authors have not addressed the concerns, and the authors do not show a good bound on C for even tabular settings.

---

> ### Author Response · Authors · 2023-01-21
> **Comments on the meta review**
>
> We appreciate the comments and feedback provided by the area chair. However, we strongly **disagree** with the assertion that "the biggest concern" is the constant in the regret bound, and thus the reasons for supporting the final decision.
>
> > The biggest concern from the reviewers is the constant in the regret bound. The authors failed to show that C is bounded for even the tabular setting, whereas previous results of PSRL usually have bounds on this.
>
> 1. In our initial submission, we had already included a remark on how to bound the constant $C\leq d$ in the tabular case with linear value functions.
>
> 2. Additionally, we have presented results on the tabular case where we proved that $C\leq |\Pi|$.
>
> 3. During the rebuttal period, we've restated these results several times (from November to January) in [link 1](https://openreview.net/forum?id=jwgnijhdF3V&noteId=XKjCyuebuu), [link 2](https://openreview.net/forum?id=jwgnijhdF3V&noteId=rFJLWMnnm1L), [link 3](https://openreview.net/forum?id=jwgnijhdF3V&noteId=0_sh5puuV7H), [link 4](https://openreview.net/forum?id=jwgnijhdF3V&noteId=zZlC-dg9Ck) and [link 5](https://openreview.net/forum?id=jwgnijhdF3V&noteId=v78Ml3BataT). But unfortunately, all these seems are overlooked.
>
> We understand the demanding workload and tight time frame for decision-making in the role of chairing papers. Nonetheless, we would like to extend our gratitude to all reviewers for your efforts in reviewing our paper and helping us improve it.